# All-optical control of second-harmonic generation in β-BaB₂O₄ via coherent, terahertz-driven acentric lattice displacement

**Flavio Giorgianni** [1,2] ✉, **Nicola Colonna** [2], **Gabriel Nagamine** [1],
**Leonie Spitz** [2], **Guy Matmon** [2], **Alexandre Trisorio**[2], **Nicolas Forget**[3],
**Carlo Vicario**[2] **& Adrian L. Cavalieri**[1,2]

Dynamical control of the nonlinear optical properties of solids – with light itself – will be essential for future ultrafast photonic technologies. Previously, methods to modulate nonlinear processes including second-harmonic generation (SHG) have relied primarily on non-resonant light-matter interaction or photo-generation of hot electrons in nanoscale materials. However, these approaches are typically constrained by limited interaction lengths and the initial frequency conversion is relatively weak under equilibrium conditions. Here, a ~30% modulation of efficient phase-matched SHG in bulk beta-barium borate (β-BaB₂O₄) is achieved through transient lattice deformation by intense terahertz (THz) pulses that are tuned to resonance with an infrared-active phonon mode. The effect originates from modification of the index of refraction ellipsoid and the corresponding nonlinear phase-matching conditions, rather than from direct modulation of the nonlinear susceptibility through THz-mediated $\chi^{(3)}$ processes. This mechanism, of resonant selective lattice excitation, points toward novel THz-control schemes to tune the nonlinear optical response in materials.

Nonlinear optical frequency conversion—including second harmonic generation (SHG)—greatly extends the spectral range of conventional lasers, underpinning key scientific applications across a wide range of fields from high-resolution microscopy to precision optical metrology to quantum information science[1–6]. For communications and technological applications, integrated photonic circuits have emerged[7–10], in which the nonlinear optical response can be precisely modulated. Control over optical nonlinearities, including SHG, using applied electric fields has now been demonstrated in optical crystals[11], polymers[12], semiconductors[13–15], atomic-layer systems and heterostructures[16,17], and artificial photonic structures[18,19]. However, modulation of the nonlinear optical response with this approach is fundamentally limited by the speed of conventional electronics to GHz modulation rates, creating a functional barrier to the next generation of optoelectronic devices. In principle, orders-of-magnitude higher speeds can be achieved through all-optical approaches, which would enable ultrafast data transfer and real-time optical signal processing.

Indeed, ultrafast, all-optical SHG modulation has been demonstrated in nanostructures and two-dimensional materials[20–23]. However, these achievements appear impractical for applications in solid-state devices, as the modulation generally relies on either photo-excitation of hot electrons[22,24] with significant collateral heating, or non-resonant light-matter interaction[25–29], which is limited by material damage thresholds. Further, while high contrast has been achieved,

[1]Institute of Applied Physics, University of Bern, CH-3012 Bern, Switzerland. [2]Paul Scherrer Institute, CH-5232 Villigen-PSI, Switzerland. [3]Université Côte d'Azur, CNRS, Institut de Physique de Nice (INPHYNI), Nice, France. ✉e-mail: flavio.giorgianni@unibe.ch

the absolute SHG signals themselves remain relatively weak due to limited interaction lengths. On the other hand, while bulk three-dimensional materials can deliver strong SHG signals suitable for applications, high-contrast, ultrafast, all-optical modulation has not yet been reported.

In this work, high-contrast modulation of strong SHG is achieved by resonant excitation of the β-barium borate (β-BaB$_2$O$_4$, BBO) crystal lattice using intense terahertz (THz) pulses tuned to an infrared (IR)-active phonon mode. Theoretical considerations − based on group theory and ab-initio calculations[30–32] − indicate that both the linear properties and the second-order optical nonlinearity in BBO originate from localized electronic density within the B$_3$O$_6$ anionic rings in the crystal unit cell. Therefore, displacement of the anionic ring can be predicted to significantly influence SHG and targeted displacements might lead to strong, controlled modulation of the SHG signal. While this parameter cannot be selectively accessed with conventional stimuli (e.g., strain, static electric fields), selective displacement can be achieved with intense THz pulses, resonantly coupled to IR-active phonons, as illustrated in Fig. 1a. With this approach, THz pulses can be expected to enable ultrafast control over the linear and nonlinear optical properties of BBO, enabling efficient modulation of the SHG process.

The potential for control by THz-driven resonant excitation is investigated here through the SHG of a femtosecond NIR probe laser. By measuring the time-resolved SHG efficiency as a function of the fundamental input wave polarization and delay with respect to the THz excitation, the modulated SHG signal yields the overall out-of-equilibrium nonlinear optical response. And importantly, under appropriate input conditions, high-contrast SHG modulation exceeding 30% can be reached.

Macroscopically, this effect can be explained by an electro-optic, THz-induced rotation of the principal optical axes (perpendicular to the direction of propagation), which alters the SHG phase matching conditions. Density functional perturbation theory (DFPT) calculations validated by the experiment can then be used to obtain an accurate estimate of the frequency dispersion of the electro-optic coefficient in the THz range, as well as the phonon normal-mode amplitude of 0.67 Å(amu)$^{(1/2)}$, which displaces the B$_3$O$_6$ anionic unit from its equilibrium position. This quantitative link between the atomic-scale structural dynamics and macroscopic nonlinear optical response may facilitate engineering of materials with specific transient nonlinear optical responses in the future.

## Results

### Experimental design and THz excitation geometry

To investigate the SHG modulation by selective lattice excitation on ultrafast timescales, intense THz pulses are used to drive the material while its linear and nonlinear response are probed with an overlapping femtosecond near-infrared (NIR) pulse (Fig. 1a, b, and Sections 1 and 2 of the Supplementary Information for details). Prior to probing the dynamics, the BBO crystal is characterized in the chosen experimental geometry in which the THz pulse is polarized along the ordinary axis (o-axis) of the BBO crystal. Steady-state optical spectroscopy (see Supplementary Fig. 3 and Section 3 of the Supplementary Information for details) yields the imaginary part of the dielectric function $\varepsilon_2(\omega)$ and the corresponding IR-active phonon modes (Fig. 1d). The dominant IR-active E-symmetry phonon mode is found at a frequency of $\omega_Q = 4.32$ THz, consistent with other recent independent characterizations by THz time-domain spectroscopy[33]. This mode is of particular interest, as DFPT calculations indicate that it leads to significant co-planar motion of anionic [B$_3$O$_6$]$^{3-}$ rings while the Ba$^{2+}$ cations will exhibit in nearly motion in the opposite direction with comparatively smaller displacements, due to their large atomic mass[31] (Fig. 1a).

Electro-optic sampling (EOS) of the driving THz pulse is used to verify spectral overlap with the E phonon mode required for targeted

excitation. In-situ EOS measurements of the THz temporal waveform $E_{THz}(t)$ yield a peak electric field strength of $E_{THz}^{peak} = 8.5$ MV/cm, and Fourier analysis reveals a spectrum heavily weighted around 4.3 THz, with additional weaker frequency components ranging up to to 10 THz (Fig. 1c, d). The spectral content ensures efficient coupling of the THz pulse to the desired E phonon mode. While there are additional IR-active phonon modes spanned by the broadband THz pulse, they have lower absorption cross-sections[33] and are not expected to contribute significantly to the driven lattice dynamics (Fig. 1e).

Following initial characterization, time-resolved experiments are performed by focusing the THz pulse into a 300 μm thick BBO crystal cut for type-I phase-matched SHG of a ~50 femtosecond NIR laser pulse at the fundamental wavelength of 800 nm. In the experiment, 20 nJ NIR pulses are focused to a spot size of 35 μm at the BBO crystal with corresponding intensity of ~8 × 10$^{10}$ W/cm$^2$, and at equilibrium a maximum SHG energy conversion efficiency of ~4% is observed.

By varying the delay between the THz pump pulse and polarization of the femtosecond NIR probe pulse, the influence of the phonon dynamics on the SHG response is investigated in the time domain as function of polarization. A key observable is the time-dependent, normalized relative strength of the SHG intensity, $\Delta I_{rel} = \Delta I_{SH}/I_{SH,0}$. Here, $\Delta I_{SH} = I_{SH,THz} - I_{SH,0}$, $I_{SH,THz}$, and the equilibrium SHG intensity, $I_{SH,0}$. By convention, time-zero occurs when the peak of the femtosecond NIR pulse coincides with the maximum of the THz electric field. Under optimized input polarization (detuned from peak conversion efficiency) and delay the THz pulse induces a modulation in the relative SHG strength exceeding 30% (Fig. 1f), whereas no significant THz-induced modulation is observed in the NIR probe intensity (Supplementary Fig. 2).

The time-dependent SHG exhibits a damped oscillatory behavior persisting for over 4 ps, with Fourier analysis revealing a distinct peak slightly below the phonon frequency, $\omega_Q$, coincident with the positive maximum of real part of the dielectric function, $\varepsilon_1$, see Fig. 1g. Tuning the spectrum of the THz driving field with spectral filters to above and below the phonon frequency $\omega_Q$. When the spectral components of the driving THz field are above $\omega_Q$, the SHG modulation is suppressed. Similarly, when the THz cut-off frequency is below $\omega_Q$, modulation of the SHG is also strongly suppressed and can be attributed to weak excitation of lower frequency phonon modes (Fig. 2a).

The phase dependence of the SHG modulation is further investigated by returning to the nominal THz pulse (tuned to resonance) and inverting its polarity. A corresponding inversion in the relative SHG intensity is observed, indicating that the dynamics are resonantly driven and phase-locked to the THz field (Fig. 2b). Additionally, scaling the THz field strength results in a linearly dependent effect, suggesting that the SHG modulation is directly proportional to the phonon amplitude, which is itself linearly dependent on the THz driving field strength[34–36] (Fig. 2c).

### Polarisation dependence

At equilibrium, the expression for the type-I SHG in the undepleted pump approximation in BBO of length $L$ by an electromagnetic wave of intensity $I$, frequency $\omega$, and polarization angle $\alpha$ is given by (see Supplementary Information Section 4):

$$I_{SH,0}(\alpha) = \frac{2\omega^2 d_{NL}^2 L^2}{n_e n_o^2 c^3 \epsilon_0} \left( \frac{\sin\left(\frac{1}{2}\Delta kL\right)}{\frac{1}{2}\Delta kL} \right)^2 \cos^4(\alpha) I^2(\omega), \qquad (1)$$

where $n_o(\omega)$ is the ordinary refractive index at the fundamental ($\omega$) and $n_e(2\omega)$ is the extraordinary refractive index at second harmonic ($2\omega$), $d_{NL}$ is the nonlinear optical coefficient, and $\Delta k$ represents the momentum mismatch between the fundamental and second harmonic waves ($\Delta k = 0$ in perfect phase matching conditions). The polarization angle $\alpha$ defines the orientation of the fundamental wave with respect

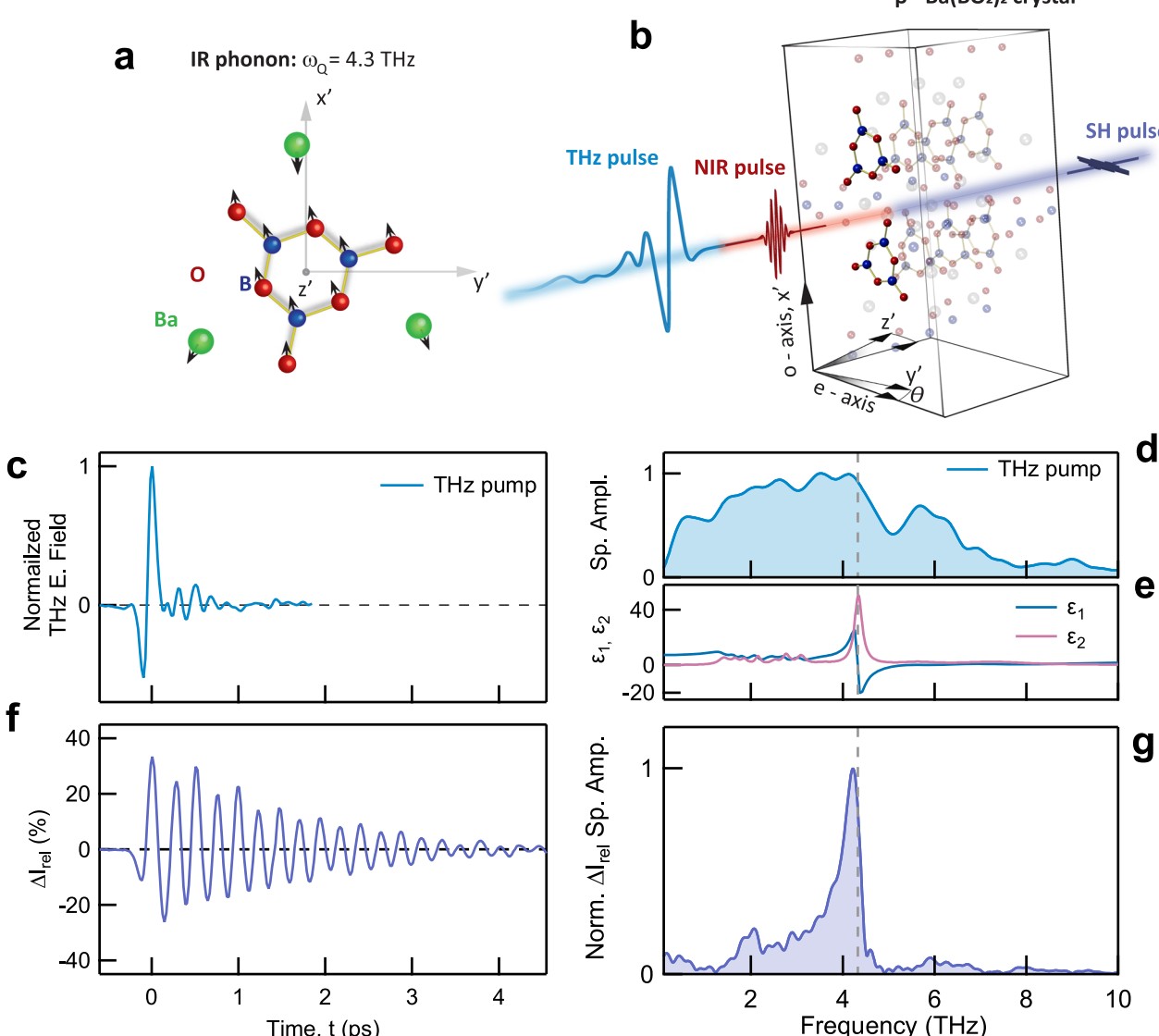

**Fig. 1 | THz-driven optical SHG modulation dynamics of β-BaB₂O₄. a** Illustration of the planar $B_3O_6$ anionic unit surrounded by Ba atoms within the unit cell of β-$BaB_2O_4$ in the crystallographic coordinates $x'$ and $y'$, along with the phonon displacement vectors projected in the crystallographic $x'$-$y'$ plane associated with the IR-active mode at $\omega_Q = 4.3\,THz$. Phonon displacement vectors have been calculated using density functional theory. **b** Schematic of the experimental setup. The THz pulse interacts with the nonlinear crystal while the SHG response is probed with a temporally synchronized ~50 femtosecond NIR pulse (FF beam) converted in a SH pulse. Arrows define the orientation of the ordinary and extraordinary axes with respect to the crystallographic axes $x'$, $y'$, and c ($\theta = 29.3°$). **c** Temporal waveform of the THz electric field and **d** corresponding normalized Fourier spectral amplitude (Sp. Ampl.). **e** Real and imaginary components ($\varepsilon_1, \varepsilon_2$) of the dielectric function of BBO (β-$BaB_2O_4$) measured using Fourier Transform Infrared (FTIR) spectroscopy. **f** THz-driven SH intensity modulation $\varepsilon_2(to-be-removed)$ dynamics (blue curve). **g** Normalized Fourier spectral amplitude (Sp. Ampl.) of the SH modulation dynamics is shown in **f**.

to the ordinary axis, as illustrated in Fig. 3a. According to Eq. (1), and consistent with equilibrium experimental observations, the SHG intensity $I_{SH,0}(\alpha)$ exhibits a cosine-to-the-fourth power dependence on input polarization, resulting in a twofold symmetric angular distribution with maximum intensity at $\alpha = 0$ and $\alpha = 180°$ (Fig. 3b).

Out-of-equilibrium, the SHG can in principle be modulated by two distinct effects. The dynamics can result from a time-dependent variation in the second-order nonlinear coefficient, which is proportional to the driving THz field and can be considered effectively as a third-order $\chi^{(3)}$ effect. Notably, under certain circumstances, this $\chi^{(3)}$-driven effect has been used to describe THz-Field Induced Second Harmonic Generation (TFISH)[37–40]. Alternatively, the dynamics can result from phonon-mediated electro-optic effects in the linear index of refraction (a $\chi^{(2)}$ effect involving THz and phonon frequencies)–and subsequent

modification of the phase-matching conditions ($\Delta k$), affecting the SHG efficiency, without inducing a time-dependent $\chi^{(2)}$ or $\chi^{(3)}$ effect, *through the second-order nonlinear coefficient involving the fundamental field frequency* (Sections 5 and 6 of the Supplementary Information). This mechanism has been described as a cascade of $\chi^{(2)}$ processes[26,41]. Under strong-field THz excitation, both $\chi^{(2)}$ and $\chi^{(3)}$ effects are always present; however, it can be expected that only one of the effects dominates the nonlinear response depending on the specific equilibrium phase-matching conditions.

To identify the origin of the observed dynamics, the NIR probe pulse polarization is scanned at a time delay of $t = 2.4$ ps - corresponding to a local maximum of the SHG dynamics. Notably, the angular dependence of the measured SHG modulation, $\Delta I_{SH}(\alpha) = I_{SH,THz}(\alpha) - I_{SH,0}(\alpha)$, now exhibits a fourfold symmetry, with

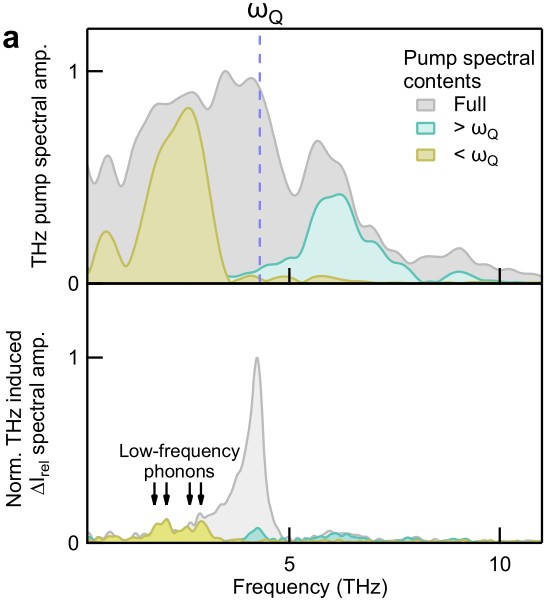

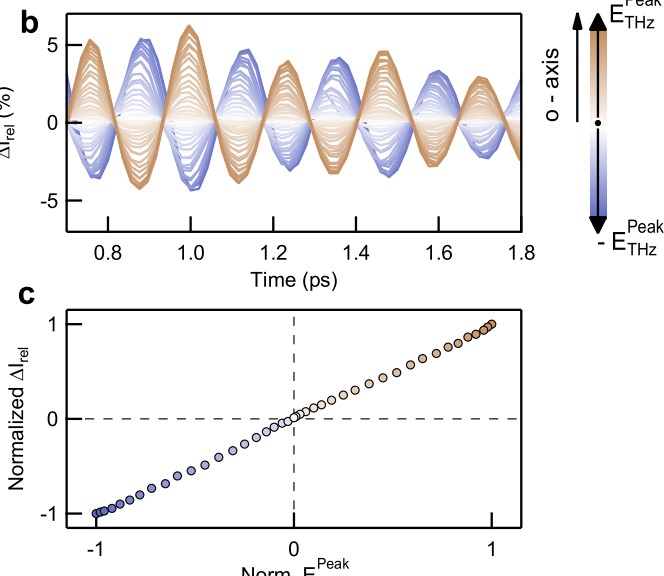

**Fig. 2 | Phonon-induced SH intensity modulation dynamics of β-BaB₂O₄. a** THz excitation of the BBO IR-active phonon modes. On resonance, a pronounced spectral amplitude (amp.) response is observed (grey curves). For comparison, low and high frequency off-resonant THz excitation (yellow and green curves) leads to a significantly reduced modulation of the second harmonic intensity. **b**, **c** The amplitude of the SH intensity modulation is directly proportional to the resonant peak field $E_{THz}^{peak}$ of the THz pump and is inverted upon reversing the polarity of the applied THz field.

the two lobes peaking at α ≈ 30° and α ≈ 330° having a positive modulation ($\Delta I_{SH}(\alpha) > 0$), corresponding to an increase in SHG intensity (Fig. 3) and vice-versa for the lobes observed at α ≈ 150° and α ≈ 210°. Further, as the temporal delay between the THz and NIR pulses is scanned over one oscillation cycle, the sign of $\Delta I_{SH}(\alpha)$ is inverted in all lobes, passing through a common zero-crossing where no modulation is observed ($\Delta I_{SH}(\alpha) = 0$), as shown in Fig. 3d.

Importantly, this angular and temporal dependence of the SHG signal in the type-I phase-matched geometry is incompatible with a TFISH interpretation involving an effective $\chi^{(3)}$-effect resulting from a THz-induced modulation of the second-order $\chi^{(2)}$ susceptibility. Evidence of this inconsistency is obtained from comparative measurements of SHG modulation in BBO where the phase-matching conditions are not met (see Supplementary Fig. 5). Here, due to the limited coherence length, the $\chi^{(3)}$ effects to be isolated, as electro-optically induced changes in the index of refraction ellipsoid are negligible in the overall SHG response. In the phase-mismatched conditions, the $\chi^{(3)}$-effects dominate, resulting in a polarization dependence in the SHG modulation with features fundamentally different from those observed in the phase-matched case. Additionally, the equilibrium signal strength and magnitude of the modulated intensity are orders of magnitude weaker (see Sections 7 and 8 of the Supplementary Information for full explanation). Therefore, in the current experiment, the strength of the effect and the fourfold symmetry leads to the conclusion that the dynamic effect is dominated by the phonon-mediated alteration of the type I phase-matching conditions—a cascaded 2nd-order effect.

**Analytical model of the cascaded $\chi^{(2)}$ dynamics**

Resonant excitation of the $E$ phonon mode by the THz pulse displaces the $(B_3O_6)^{3-}$ anionic rings far from equilibrium positions (Fig. 1a), thereby inducing an electro-optic modulation of the refractive index along the crystal's principal axes ($x', y', z'$; see Fig. 1a). Under equilibrium conditions, BBO is a negative uniaxial crystal ($n_{x'} = n_{y'} \neq n_{z'}$), but the phonon excitation breaks its axial symmetry, leading to biaxial behavior where all three refractive indices become distinct ($n_x \neq n_y \neq n_z$) – see Section 5 of the Supplementary Information for details. The

dynamical refractive indices along the $x$ and $y$ axes in the crystal-frame become: $n_{x'}^{THz} = n_{x'} + \delta n$ and $n_{y'}^{THz} = n_{y'} - \delta n$, where the magnitude of the phonon-induced refractive index modulation is:

$$\delta n = n_{x',y}^3 r_{22} Q / 2 \qquad (2)$$

Here, $Q$ is the phonon amplitude and $r_{22}$ is the phonon-optic coupling coefficient, relating the modulation of the refractive index to the phonon amplitude. To interpret the index of refraction ellipsoid in the context of phase matching, which governs the efficiency of the SHG $\chi^{(2)}(2\omega, \omega, \omega)$-process, a projection normal to the direction of propagation in the laboratory frame is taken from which the indices along the ordinary and extraordinary axes, known as the principal axes, are found.

In the experiment, the BBO crystal was cut for optimal type-I phase matching at equilibrium (SHG at 800 nm), with its crystal axes oriented at $\theta = 29.3°$ and $\varphi = 90°$ (following the conventions in ref. 42). The cut angle defines the direction of propagation in the crystal system and ensures that the refractive index of the ordinary fundamental wave ($n_o$) matches that of the extraordinary second harmonic wave ($n_e$). In the dynamical case, the phonon-induced modulation of the index of refraction ellipsoid ($\delta n$ in the crystal coordinate system) leads to a rotation of the ordinary and extraordinary axes in the laboratory frame by an angle $\gamma = \kappa(\theta) r_{22} Q$, as illustrated in Fig. 4a. Here, $\kappa(\theta)$ is a geometrical parameter that depends on the crystal orientation $\theta$.

While the resonant phonon excitation rotates the ordinary and extraordinary axes, impacting the phase-matching conditions in the SHG process, the magnitudes of the ordinary and extraordinary refractive indices—apart from a third-order correction—remain nearly unchanged. (Derivations of the modulation in the refractive index $\delta n$, the effective rotation angle $\gamma$ of the principal axes, and all relevant details are given in Sections 5 and 6 of Supplementary Information).

To quantitatively model the dynamics, it is also essential to consider the spatio-temporal propagation of the phonon inside the BBO crystal. The phonon amplitude decays exponentially along the direction of propagation. This spatial dependence of the phonon amplitude

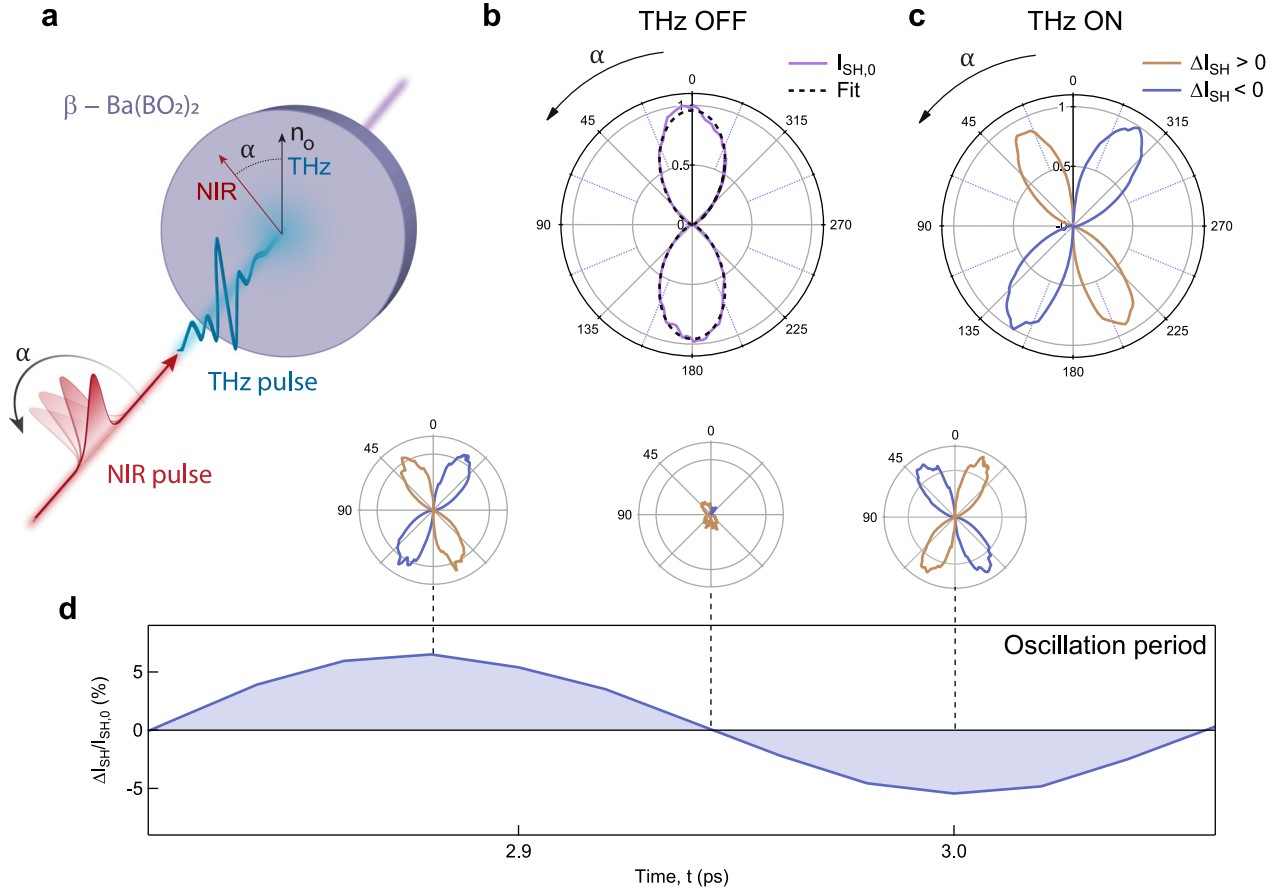

**Fig. 3 | Polarization resolved THz-induced SHG modulation. a** The angle $\alpha$ between the NIR pulse polarization and the BBO ordinary principal axis is varied while the THz field is kept parallel to the ordinary axis. **b** The SHG intensity as a function of $\alpha$ is fit by $\cos^4\alpha$ curve (black dashed line), which is the expected angular dependence in type-I SHG. **c** THz-induced SHG modulation $\Delta I_{SH}(\alpha)$ versus $\alpha$ at fixed THz-NIR delay time of 2.4 ps. The orange solid curve denotes an increase of the SH intensity ($\Delta I_{SH}(\alpha) > 0$) while the blue line a reduction ($\Delta I_{SH}(\alpha) < 0$). **d** The positive and negative modulation quadrants are interchanged by delaying the THz-NIR by one phonon half period.

can be described by $Q(z) = Q_0 e^{-\frac{z}{\delta_{ph}}}$, where $Q_0$ is the initial phonon amplitude at the crystal surface, and $\delta_{ph}$ is its penetration depth. By accounting for the phonon-induced rotation of the ordinary and extraordinary optical axes, as well as the penetration depth of the phonon, the variation in SHG intensity as a function of the input polarization angle $\alpha$ (at fixed THz-NIR delay) is then given by:

$$\Delta I_{SH}(\alpha) = 4 I_{SH,0}(0) \gamma_0 \frac{\Lambda(\delta_{ph}, L)}{L} \cos^3\alpha \sin\alpha, \qquad (3)$$

where $I_{SH,0}(0)$ is the equilibrium SH intensity at α=0, and $\gamma_0 = \kappa(\theta) r_{22} Q_0$ is the maximum rotation angle of the optical axes induced by the phonon amplitude $Q_0$ at the fixed delay. The fourfold dependence on the input polarization, $\sim \cos^3\alpha \sin\alpha$, is consistent with the alternating sign and angular symmetry observed experimentally, and the strength of the effect depends on the BBO thickness, $L$ and $\Lambda(\delta_{ph}, L)$, which includes the phonon penetration depth.

A key parameter is the phonon penetration depth $\delta_{ph}$, which can be estimated independently using finite-difference time-domain (FDTD) methods to calculate the spatio-temporal dynamics of the phonon amplitude. With FTIR measurements of the linear optical properties of the BBO crystal, EOS measurements of the THz driving electric field, and the phonon equation of motion, the normalized amplitude $Q(z,t)$ of the phonon at $\omega_Q$ is simulated using FDTD and shown as a function of the crystal depth and time in Fig. 4b. Integrating

the phonon amplitude in time and fitting the depth-dependent profile with an exponential decay (Fig. 4c) yields a phonon penetration depth of $\delta_{ph} = 30.2\ \mu m$.

Using $\delta_{ph}$, angularly dependent SHG measurements (at fixed time delay) for THz pulses of varying intensity are then fit with Eq. 3, keeping $I_{SH,0}(0)$ as a constant parameter. These fits yield $\gamma_0$ as a function of the THz field amplitude (Fig. 4d, e). The rotation of the ordinary and extraordinary axes, $\gamma_0$, is found to vary linearly with applied THz field amplitude, consistent with the linear dependence of the IR-active phonon amplitude $Q_0$ on the THz field strength. The value of $\gamma_0$ achieved for the maximum THz field strength in the experiment is $\gamma_{0,\ max} = 0.216\ rad$.

## Microscopic mechanisms of phonon-driven nonlinear optical modulation

The polarization-resolved measurements in conjunction with the analytical analysis yield the rotation of the optical axes due to the resonant THz-phonon excitation. This, in turn, permits quantification of the associated modulation of the refractive indices $\delta n$ in the crystal coordinate system, but does not immediately allow for independent retrieval of the electro-optic coefficient $r_{22}^E$ or a determination of the phonon displacement $Q$. At the microscopic level, the effect of the THz-induced coherent lattice vibrations on the local electronic environment within the anionic units remains hidden, as does the electro-optic coefficient $r_{22}^E$.

To access this information density functional theory (DFT) and density functional perturbation theory (DFPT) can be used in the

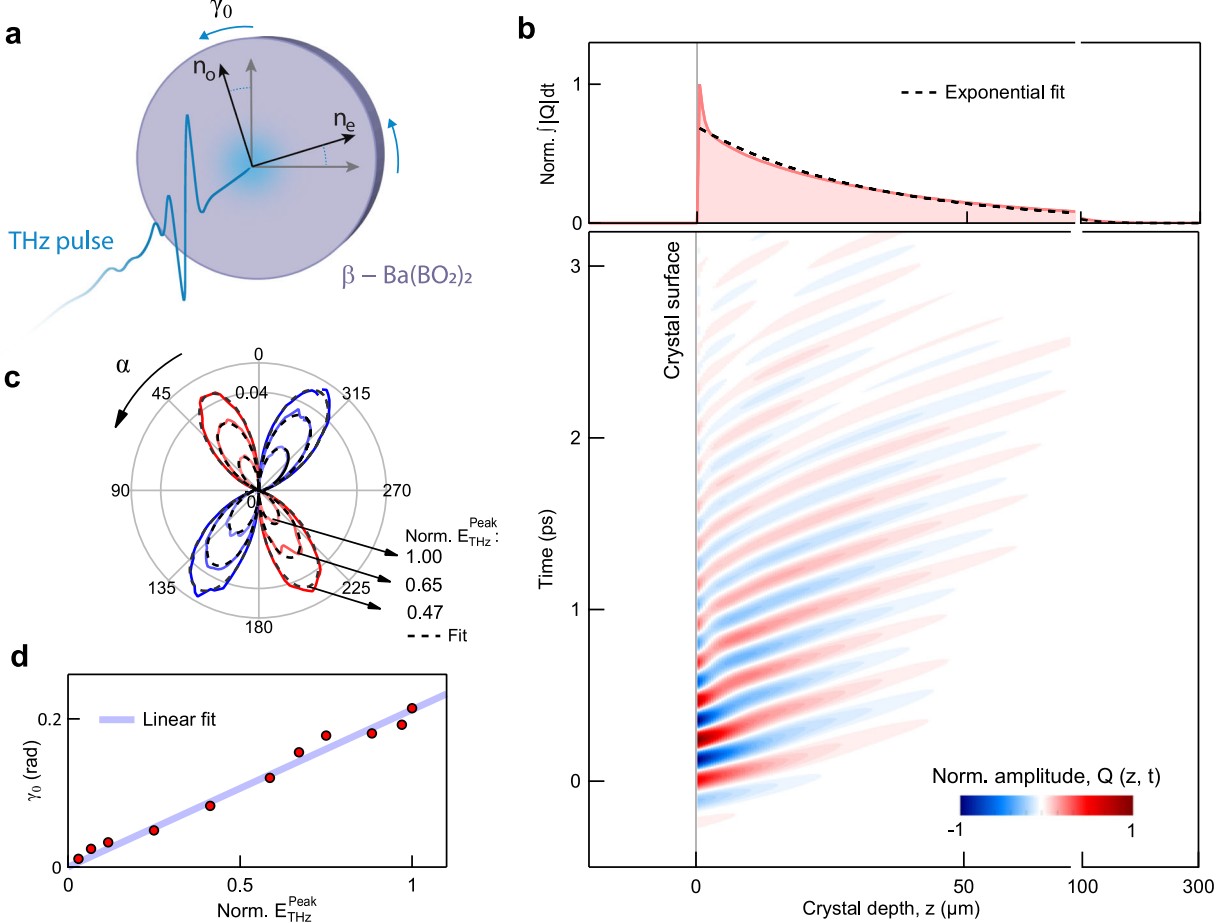

**Fig. 4 | Theoretical modeling of SHG modulation and FDTD simulation for the phonon propagation. a** The BBO principal axes with corresponding refractive indices $n_o$ and $n_e$ undergo a rotation in the laboratory frame by an angle $\gamma_0$ due to the lattice displacement Q. **b** Spatio-temporal evolution of the normalized (norm.) THz-driven phonon amplitude $Q(z,t)$ inside the bulk BBO crystal calculated using FDTD method. Top panel: FDTD results for the time-integrated phonon amplitude $Q$ as a function of depth in the crystal (red curve) and the exponential fit of the

decay, $Q(z) = Q_0 e^{-z/\delta_{ph}}$ (black dotted curve), to estimate the phonon penetration depth $\delta_{ph}$. **c** The phonon-driven rotation of the principal axes, $\gamma_0$, results in a polarization-dependent SHG modulation with $cos^3(\alpha) sin(\alpha)$ angular dependence (Eq. 2). Dashed black lines are fits (based on Eq. 3) of the measurements as a function of the normalized THz field strength at a fixed delay of 2.4 ps between NIR probe and THz pump-pulse. **d** Estimate of $\gamma_0$ as a function of peak THz pulse field strength using the fits to polarization resolved SHG modulation measurements.

frozen-phonon approximation, to calculate the electronic band structure[32] and the dielectric function corresponding to a lattice displacement $Q_0$ along the phonon coordinates. The resulting dielectric function can then be used to extract absolute values for the refractive indices, $n_{x'}^{THz}$ and $n_{y'}^{THz}$ in the $(x', y', z')$ coordinate system of the crystal as a function of the phonon displacement $Q_0$ (Fig. 5a). Starting from equilibrium ($Q_0 = 0$), both indices are found to be $n_{x'} = n_{y'} = 1.705$, in good agreement with experimental measurements and previous ab-initio calculations[32]. With a finite phonon amplitude $Q_o$, the refractive indices respond linearly as expected: $n_{x'}^{THz}$ increases, while $n_{y'}^{THz}$ decreases with increasing $Q_0$ (see Fig. 5a).

Using the DFPT-calculated dependence of $\delta n$ on $Q_0$, the phonon-optic coupling coefficient $r_{22}$ can be estimated from the relation, $r_{22} = 2\delta n(Q_0)/n_{x'}^3 Q_0$, yielding a value of $r_{22} = 0.0049 A^{-1}(amu)^{-1/2}$. This coefficient is especially relevant, as it can be used to calculate the maximum phonon amplitude at maximum THz field strength using the experimentally determined rotation of the optical axes $\gamma_{0, max}$ at the crystal surface, $Q_{0, max} = \gamma_{0, max}/(\kappa r_{22}) = 0.67 A(amu)^{1/2}$ (Fig. 5a). This value is consistent with expectations based on experiments targeting different effects in various materials that are nevertheless similar in that they also require THz excitation of large amplitude phonons. In these independent experiments, performed at X-ray free-electron

lasers, it was possible to make a direct structural determination of the phonon amplitude[35,43].

Finally, the electro-optic coefficient $r_{22}^E(\omega)$ can be deduced from phonon-optic coupling coefficient $r_{22}$ using the conventional linear relation between the THz field and $Q$ in the frequency domain, which leads to the following expression:

$$r_{22}^E(\omega) = r_{22}^E(0) \frac{\left(\omega_Q^2 - \omega^2\right)}{\left(\omega_Q^2 - \omega^2\right)^2 - (\Gamma\omega)^2}, \quad (4)$$

where $\omega_Q$ and $\Gamma$ are the phonon frequency and linewidth and the $r_{22}^E(0)$ is the DC electro-optic coefficient (see Section 9 of the Supplementary Information for details). Approaching the phonon resonance, the $r_{22}^E(\omega)$ peak value is 55 pm/V at $\omega_* = \sqrt{\omega_Q^2 - \Gamma\omega_Q} = 4.22$ THz. Notably, the spectrum of the time-dependent SHG modulation dynamics closely matches the EO coefficient, with shared maxima just below $\omega_Q$ (dashed line Fig. 5b). This frequency dependence provides further evidence that the SHG modulation arises from the electro-optic effect. Indeed, in phonon-mediated EO processes, the response is governed not by the imaginary part of the dielectric function, $\varepsilon_2(\omega)$, which peaks

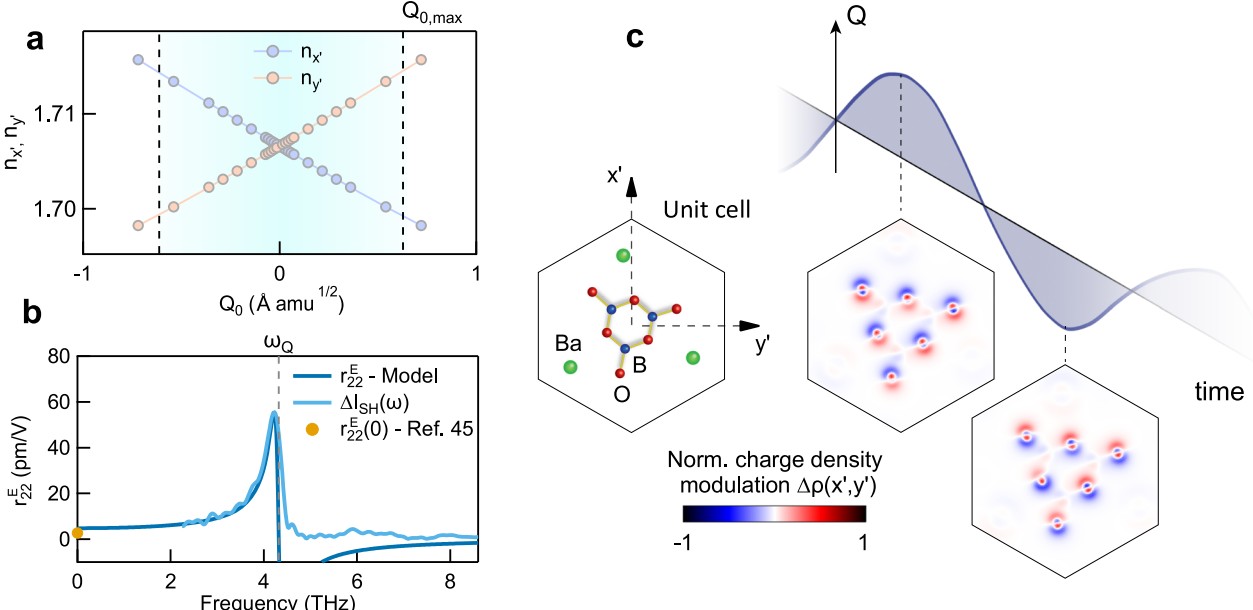

**Fig. 5 | Frozen-phonon DFPT calculation results and phononic-modulation spatial charge-density modulation. a** Computed refractive indexes $n_{x'}^{THz}$ and $n_{y'}^{THz}$ along the $x'$ and $y'$ axes of the unit cell as a function of $Q_0$. Vertical dashed lines, which delimit the shaded light-blue region, indicate the maximum amplitude estimated in the experiment, $Q_{0,\,max} \sim 0.67 A(amu)^{1/2}$ at 2.4 ps. **b** Frequency dependence of the electro-optic coefficient $r_{22}^E(\omega)$, shown together with the spectrum of the SHG-modulation dynamics for comparison. **c** Planar modulation of the normalized (norm.) charge density $\Delta\rho(x', y')$ in the unit cell of β-BaB$_2$O$_4$ induced by phonon displacement for $Q_{0,\,max}$ and $-Q_{0,\,max}$. The electronic density is significantly affected by the O atomic displacement of the B$_3$O$_6$ unit.

at the phonon frequency $\omega_Q$, but by the real part, $\varepsilon_1(\omega)$, whose maximum occurs at $\omega_*$[44].

The peak value of the vibrational electro-optic coefficient can also be estimated experimentally by combining the analytical expression for the SHG modulation with the phonon equation of motion (see Section 9 of the Supplementary Information). Using this approach and the broadband THz waveform applied in the experiment, a peak value of $r_{22}^E$ of ~140 pm/V is obtained in relatively good agreement with that obtained by DFPT (see Supplementary Fig. 7). To exclude the involvement of other phonon modes, additional experiments were performed using a narrowband THz pulse, which yields a similar value of ~170 pm/V (see Supplementary Fig. 8). The discrepancy of less than a factor of 3 between theory and experiment, which exists in both the broadband and narrowband case could therefore stem from the calibration of the absolute THz field amplitude, phonon parameters, and on the optical constants involved in the analysis—including their large frequency dispersion near the phonon resonance and dependence on the geometrical parameters in the experiment.

On the other hand, the theoretical treatment could also be slightly flawed, as extrapolating the theoretical electro-optic coefficient to the DC limit gives $r_{22}^E(0) = 6.6\,pm/V$, which is larger than the commonly reported experimental DC value of approximately 2.5 pm/V for BBO[45].

Regardless of the small discrepancy that remains, relative agreement between the experimental and theoretically determined electro-optic coefficient lends credence to the DFPT calculations, by which additional parameters can be obtained in the vicinity of the phonon resonance. In particular, the interaction between the phonon and the local electronic environment can be explored computationally. Here, DFT-calculated differential charge density maps $\Delta\rho(x', y')$ confirm that the phonon mode predominantly drives cooperative displacements of the anionic [B$_3$O$_6$]$^{3-}$ rings within the lattice plane, while the Ba$^{2+}$ cations remain largely stationary (Fig. 5c). This selective atomic motion produces a marked redistribution of charge density around the planar

[B$_3$O$_6$]$^{3-}$ units. The electronic density around the oxygen atoms is largely influenced by the phonon dynamics, suggesting that the local electronic environment of these atoms plays a central role in modulating the refractive indices $n_{x'}^{THz}$ and $n_{y'}^{THz}$.

## Discussion

Coherent lattice dynamics driven by intense THz pulses tuned to resonance with an infrared-active phonon mode have been shown to strongly modulate SHG in BBO on ultrafast timescales with the phase of the driving optical field. Analytical modeling and first-principles DFPT calculations provide both a macroscopic overview and a comprehensive description of the physics at the microscopic level.

At the atomic level, DFPT calculations indicate that the THz-driven phonons induce acentric displacements of the B$_3$O$_6$ anionic rings in the BBO unit cell. These displacements are phase-locked to the phonon dynamics (and driving field) and estimated to be as large as ~ 0.5 A$^{-1}$ (amu)$^{-1/2}$, leading to a redistribution of local charge around the oxygen atoms. The impact of the transient charge redistribution on the refractive index ellipsoid of BBO is quantified through DFPT calculations. Additionally, the phonon-mediated electro-optic rotation of the ordinary and extraordinary optical axes that governs SHG is quantified using an analytical model. The observed SHG-modulation spectrum and angular dependence are consistent with the predicted EO response, exhibiting a resonance immediately below the IR-active phonon frequency. From a macroscopic perspective, it is the link between the rotation of the optical axes—which are defined at equilibrium by the experiment geometry—and the optical phase-matching conditions that permit strong modulation of the SHG signal in the bulk material that exceeds 30%.

Excellent agreement between theoretical predictions and experimental observations shows that an approach that combines microscopic computational and macroscopic analytic treatment can be leveraged to capture the interplay between phonons and electronic states underlying nonlinear optical phenomena in solids[30]. These findings also establish DFPT as a powerful framework in which the

optical properties of out-of-equilibrium crystal structures can be accurately evaluated.

Beyond fundamental insight, these findings may impact the design of new materials with tailored transient nonlinear responses, advancing ultrafast photonics and THz technologies. The ability to dynamically modulate SHG with high efficiency and precision through lattice-driven mechanisms opens new opportunities for ultrafast optical signal processing, high-speed data communication, and nonlinear optical switching. By leveraging lattice degrees of freedom[46,47], this work introduces a resonant pathway for ultrafast, high-efficiency control of nonlinear optical responses in three-dimensional bulk materials. Further, by general application of DFPT as a predictive tool for the optical-electronic-lattice interaction, combined with analytical treatment and experiments in the future, it can be expected that a broad range of nonlinear optical responses in other materials may be optimized and tailored for specific functionalities.

## Methods

### Experimental details

Intense THz pulses were generated through optical rectification in a DSTMS organic crystal using the signal beam of an optical parametric amplifier at wavelength of 1500 nm, driven by 50 fs, 800 nm pulses at a 100 Hz repetition rate. The SHG dynamics in β-BaB$_2$O$_4$ were investigated using a time-delayed replica of the 800 nm pulses. The experimental setup is shown in Supplementary Fig. 1; further details are provided in Section 1 of the Supplementary Information.

All experiments were performed at room temperature. The sample used was a commercially available β-BaB$_2$O$_4$ single crystal (5 mm × 5 mm × 0.3 mm), oriented at $\theta = 29.3°$ and $\phi = 90°$. Crystal orientation and crystallographic quality were verified by Laue X-ray diffraction (Supplementary Fig. 4).

### FDTD spatio-temporal simulations of phonon propagation

The THz-driven phonon propagation along the crystal depth has been simulated in the time domain by computing Maxwell's equations based on FDTD method[48]. The material optical response along the o-axis at THz frequencies was modeled including the dielectric function of IR-active phonons, whose parameters have been experimentally determined using FTIR spectroscopy (see Section 3 of the Supplementary Information). The dominant feature is the phonon at $\omega_Q$, but we include additional 13 phonons for a faithful description of the optical response (phonon parameters are listed in Supplementary Table 1). For each ith phonon mode, the equation of motion in the time domain is given by[35]

$$\ddot{Q}_i + 2\Gamma_i \dot{Q}_i + \omega_{Q,i}^2 Q_i = Z E_{THz}(t) \tag{5}$$

Where $\Gamma_i$ is the damping rate, $\omega_{Q,i}$ is the phonon frequency and $Z = \omega_{Q,i}\sqrt{\epsilon_0(\varepsilon_0 - \varepsilon_\infty)}$ is the phonon effective charge, being $\varepsilon_0$ and $\varepsilon$ the dielectric constant at the low and high-frequency limit, respectively, and $\epsilon_0$ the vacuum permittivity.

Lattice equations and the electric field component from Maxwell's equations in media are related by displacement field as:

$$D = \epsilon_0 \varepsilon_\infty E_{THz} + \sum_i \omega_{Q,i}\sqrt{\epsilon_0(\varepsilon_0 - \varepsilon_\infty)} Q_i \tag{6}$$

By FDTD calculation, we numerically solve the above equation at points of the grid through leapfrog time-step scheme. In the simulation, the THz electric field $E_{THz}(t) = E_0 \cos(\omega_0 t) e^{-\frac{1}{2}\frac{t^2}{\sigma^2}}$ is a Gaussian field having a width $\sigma = 100$ fs and a center frequency $\omega_0 = 5$ THz. The plot of the spatiotemporal propagation of the phonon at $\omega_Q$ is show in Fig. 4b.

### DFPT calculations of linear and nonlinear optical properties under phonon displacement

To quantify the linear and nonlinear optical responses due to phonon displacement, first-principles calculations were performed in the framework of DFPT as implemented in the PHonon package[49] Quantum ESPRESSO[50,51], with PBE SG15 Optimized Norm-Conserving Vanderbilt pseudopotentials[52,53].

DFT and DFPT calculations were conducted within the structural unit cell [42 atoms, R3c (No. 161) space group] of the BBO crystal at room temperature[54]. For the Brillouin zone sampling, $4 \times 4 \times 4$ k-point grid and a plane-wave energy cutoff of 80 Ry have been used and the structural parameters have been relaxed until the force acting on each atom was less than 0.0025 eV/Å and the pressure was below 0.5 kbar. Through phonon energies and corresponding displacement vectors at the Brillouin zone center, the macroscopic dielectric tensor, and Born effective charges were computed.

The refractive index ellipsoid was computed as the square-root of the macroscopic dielectric tensor, calculated with DFPT in the long-wavelength limit ($q = 0$). This is valid in non-absorbing materials, in which the refractive index dispersion is small, as in the case of BBO in the optical region.

At zero displacement along the phonon coordinates ($Q_0 = 0$), the DFPT-calculated dielectric tensor in the crystal frame is:

$$\xi_0 = \begin{pmatrix} \xi_{x'x'}^0 & 0 & 0 \\ 0 & \xi_{x'x'}^0 & 0 \\ 0 & 0 & \xi_{z'z'}^0 \end{pmatrix} \tag{7}$$

which corresponds to a refractive $n_{x'} = \sqrt{\xi_{x'x'}} = 1.705$ and $n_{z'} = \sqrt{\xi_{z'z'}} = 1.705$. Previous DFPT studies have reported a refractive index of $n_{x'} = n_{y'} = 1.692$ at 800 nm, see ref. 32, which is within 0.5% of our DFPT-calculated value. This close agreement validates the use of the long-wavelength approximation in our approach. Additionally, the small discrepancy might be due to the fact our DFPT method, ref. 49, includes local-field effects and goes beyond the independent-particle approximation employed in ref. 32.

Experimentally, the refractive index has been measured to be $n_{x'} = 1.66$, which is within 3% of our calculated results.

The dielectric tensor as a function of the phonon amplitude was computed with the same approach but assuming structures displaced along the relevant IR-active phonon mode displacement vectors.

The dielectric tensor as a function of the phonon amplitude has the following form:

$$\xi_Q = \begin{pmatrix} \xi_{x'x'} & \xi_{x'y'} & 0 \\ \xi_{x'y'} & \xi_{x'x'} & 0 \\ 0 & 0 & \xi_{zz} \end{pmatrix} \xrightarrow{R(\frac{\pi}{2})} \begin{pmatrix} \widetilde{\xi}_{x'x'} & 0 & 0 \\ 0 & \widetilde{\xi}_{x'x'} & 0 \\ 0 & 0 & \xi_{z'z'} \end{pmatrix} \tag{8}$$

which diagonalized with a $x' - y'$ plane $\frac{\pi}{2}$ rotation around the z' axis, as in the analytical model (see Section 6 of the Supplementary Information). For the dielectric tensor in Eq. 7, the refractive indices as a function of phonon amplitude were calculated.

For the electronic density calculations in Fig. 4b, a plane-wave kinetic energy cutoff of 320 Ry has been used. To evaluate the nonlinear optical coefficients of BBO (see Supplementary Fig. 6) finite-field approach was used[55]. In this method, the electric field is explicitly included in the DFT calculations, and the nonlinear coefficients are obtained numerically by computing the second derivative of the polarization with respect to the applied electric field.

## Data availability

The data supporting the findings of this study, including those presented in the Supplementary Information, are available in Figshare at https://doi.org/10.6084/m9.figshare.31254352.

## Code availability

The DFT and DFPT calculations presented in this study are available on Materials Cloud at https://doi.org/10.24435/materialscloud:k3-ns.

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

## Acknowledgements

The authors thank the PSI-LNO GL group for support during the experiments and R. Li for helpful discussions. This work was supported by the Swiss National Science Foundation (SNSF) under grant number 10001644 and SNSF Spark grant number 221173, and by the French government through the UCA J.E.D.I. Investments in the Future project managed by the National Research Agency (ANR) under reference number ANR-15-IDEX-01.

## Author contributions

F.G. and C.V. conceived the project and designed the experiment. F.G., C.V., and G.M. performed the THz spectroscopy measurements. F.G., C.V., A.T., G.N., and L.S. carried out the THz experiments. F.G. performed the FDTD simulations. N.C. carried out the DFT and DFPT calculations. N.F., N.C., and F.G. developed the analytical model and interpreted the data. F.G. prepared the figures. F.G., A.L.C., and C.V. wrote the manuscript with input from all authors. All authors discussed the results.

## Competing interests

The authors declare no competing interests.
