## [Transparent Peer Review file · Nature Communications]

All-optical control of second-harmonic generation in β -BaB₂O₄ via coherent, terahertz-driven acentric lattice displacement

Corresponding Author: Dr Flavio Giorgianni

Version 0:

Reviewer comments:

Reviewer #1

(Remarks to the Author)

Present manuscript reports the observation of THz field modulated SHG effect in the BBO crystal that has potential applications for fast optical control in photonic technologies. The modulated response was convincible attributed to the THz electric field induced birefringence, i.e. electro-optic effect, which in turn modulates the SHG phase matching conditions through the phase mismatch Δk . Since the local field of THz waves could be very high and close to resonance with certain vibration mode that make major contributions to the EO coefficients, the enhancement of the EO effect could be substantial. These findings are novel and have strong tendency towards applications hence deserve to be published in NC. Before being accepted, there are following concerns or suggestions to the authors:

1. Estimation of the THz field, a number of 8.5 MV/cm was mentioned in the text. Presumably that's for the whole spectral range, how much it will be around 4.3 THz (e.g., ± 0.1 THz) it could be potentially important to estimate the EO coefficient close to phonon resonance.
2. In fig. 1d and f and throughout the text, the authors seemed to believe that the spectrum of modulated SHG signal has a peak at exact the phonon frequency of 4.3 THz. However, at the exact resonance the penetration depth of the THz field could be very limited (perhaps ~ 1 μm since the absorption coefficient is normally on the order of 10^4 cm^{-1} at resonance). And above the resonance, contribution from the phonon mode will go negative to the EO effect. Actually, it looks like that the peak position of the SHG spectrum (Fig. d and f) is slightly below that of the peak in ϵ_2 , please confirm that.
3. Related to 2, the ϵ_2 curve and its peak positions were obtained by fitting the FTIR absorption spectrum of thin BBO crystal (Fig. S3). However, from 4 to 8 THz the transmittance is practically zero. How much reliable will be the phonon frequencies spectral weights determined this way in that range (e.g. 4.3 THz-144.83 cm^{-1} , and 188.78, 245.15 cm^{-1}).
4. In the text, line 268, the refractive indices by DFT calculation were given as $n_x = n_y = 1.705$, yet in the in line 374, $n_x = 1.76$ and $n_y = 1.76$ were listed. At what wavelength of those values are referred to, it is about 1.66 at 800 nm from measured values.
5. The vibrational electro-optic coefficient r_{22} was estimated to be 0.0014 $\text{\AA}^{-1}(\text{amu})^{-1/2}$, which is not in the standard unit. It must be converted to the SI unit (m/V) for comparison to the electronic contributions or the contributions from all the phonons at off-resonance frequencies.

Reviewer #2

(Remarks to the Author)

This paper reports on a TFISH experiment, in which the TeraHertz modulation of the second order optical nonlinearity of the material is achieved through excitation of an infrared active phonon, which can potentially provide greater response than nonlinear excitation.

The work is potentially interesting and although it is not so new (there has been work in phononics probed with SHG, for example in the study of ferroelectrics or cuprates). Nevertheless, the nonlinear optical implications have not been discussed in detail.

a suitably revised version of this manuscript would in principle be appropriate for Nature Communications.

Yet, there are some major flaws in the present discussions that the authors must address. I must say that the authors do not strike me as having a particularly deep understanding of the microscopic physics behind their experiment. I suggest that the authors think about this work more deeply and consult with experts in phononics, Nonlinear optics and related fields.

I list some of the most important flaws I have found when studying this manuscript.

1) The material is not centrosymmetric, hence the mode cannot be of E_g symmetry. In a non-centrosymmetric medium one cannot separate modes in even and odd, since they are all infrared and Raman active. Parenthetically, if the mode were of E_g symmetry it would not be infrared active because it would be even. I think that the authors should take a look at their nomenclature and correct accordingly.

2) The phonon mode being driven is being displaced by the electric field. Hence, the relevant quantity is a Polarization P , which is proportional to χ and hence to ϵ_1 (not ϵ_2). The figure compares the response to ϵ_2 , which is a measure of coherent dissipation, and not of the induced phonon dipole. They should plot the loss function of the material ($1/\text{Im}(\epsilon_1)$) and compare the observed oscillation frequencies to existing phonons. Please correct accordingly.

3) More importantly. What are the authors measuring in their experiment? There are at least two contributions to the second harmonic signal when modulated by an IR phonon QIR. One contribution, which is a homodyned response, will be proportional to the square of the SH field and therefore one expects oscillations at twice the fundamental frequency. There will also be a heterodyne contribution, which will presumably rely on the interference between static SHG from the BBO crystal and the induced lattice oscillations. Have the authors studied the effect in detail and are they sure which made they are looking at? Oscillations at 4.3 THz could be a homodyned SHG from a 2.15 THz phonon or heterodyne oscillations from a 4.3 THz phonon. The best way to solve this issue would be to add a local oscillator to make sure that the heterodyned component dominates (see for example Mankowsky et al. PRL 118, 197601 (2017))

4) Results of figure 4. I was confused in reading about this part of the paper. If the mode is an E mode, it means that it is doubly degenerate and I imagine that there will be two sets of mode displacements (let's call them E_1 and E_2) which will respond to the THz drive. If I look at figure 1 I imagine that a second set of vibrations will be found at some angle. Can the authors clarify this aspect? Maybe this is already contained in their analysis and hence this simply needs to be made clearer in the text.

5) The authors should maybe mention that this effect is in fact a third order ($\chi^{(3)}$) effect, in which two photons from the mid-infrared pulse are mixed with one photon from the THz modulator. Indeed, the modulated SHG will be upshifted and downshifted by one phonon, which may be important for the phase matching conditions. In this sense, the propagation may have to include polariton equations, since on and off resonant excitation of a phonon will lead to a modulating field that propagates at vastly different group and phase velocities.

In summary, this is a paper with some potential, but in its current form, the paper is unpublishable. I am happy to help cleaning this paper up, but the manuscript requires a lot of work.

Version 1:

Reviewer comments:

Reviewer #1

(Remarks to the Author)

After carefully reading their revised manuscript, replies to comments and SI, it is found that all my previous concerns were considered and corresponding revisions were made satisfactorily. Therefore I have no objection to accept present manuscript.

ps. there is a missing reference R5 in the SI, please update it.

Reviewer #2

(Remarks to the Author)

The authors have done a good job and convincingly rebutted the vast majority of my concerns. There is no point in litigating a few things that I am still not 100% clear about. These are marginal in the context of the claims made here and would add unnecessary delays.

I am happy for this paper to be published as is and I would like to congratulate the authors for the nice results and to thank them for their comprehensive clarifications.

Reply to review

Reply to review report 1:

We thank the reviewer for time spent evaluating our manuscript, for constructive feedback, and for their provisional recommendation for publication in *Nature Communications*. We are pleased that the reviewer finds our observations of all-optical control of phase-matched SHG in BBO to be both novel and relevant for future photonics applications.

In response to both reports, we have revised the manuscript substantially, which we believe has improved its clarity and overall quality. Specifically in response to the first report, we (i) estimated the magnitude of the vibrational electro-optic coefficient in SI units experimentally using broadband and narrowband THz pumping, which required additional measurements, and computationally via density-functional perturbation theory (DFPT); (ii) compared phonon frequencies obtained from FTIR with literature values from THz-TDS and Raman spectroscopy; (iii) provided additional details on the refractive-index calculations performed with DFPT; and (iv) compared the frequency-dependent features of the SHG modulation dynamics with the frequency dependence of the vibrational electro-optic coefficient.

We believe that these revisions and additional information, included in direct response to the referee's suggestions, substantially clarify and strengthen our findings. Below, we provide a point-by-point response to the reviewer's comments (in italics) and list the corresponding changes made in the manuscript.

1. THz field strength:

“Estimation of the THz field, a number of 8.5 MV/cm was mentioned in the text. Presumably that's for the whole spectral range, how much it will be around 4.3 THz (e.g., ± 0.1 THz), it could be potentially important to estimate the EO coefficient close to phonon resonance.”

Presumably, due to sharp localization of the phonon response in the frequency domain around ω_Q , frequency components in a broadband THz pulse far from ω_Q will contribute negligibly to the phonon dynamics. Therefore, an “effective” resonant field E_{THz}^{Eff} can be defined as the portion of the THz electric field whose spectrum overlaps with the phonon response that would also produce an equivalent phonon motion as the full broadband THz drive.

We can estimate this effective field by first transmitting the full THz pulse (measured by EO sampling) into the BBO crystal using the Fresnel transmission coefficients (calculated from the FTIR-measured refractive index). Then the Fourier transform of the THz pulse in the time domain is evaluated and filtered using a gaussian filter with 0.2 THz FWHM). These frequency components are then converted back to the time domain yielding the effective THz field. Following this procedure, the peak field strength of the effective THz field, with spectral components centered around the phonon frequency, was determined to be ~ 500 kV/cm.

We note that our calculation of the electro-optic coefficient, as described in Section S9 of the Supplementary Information and reproduced in Section 5 (“Estimation of the electro-optic coefficient in SI units”) of this report, does not require the explicit isolation of individual Fourier components of the THz pulse, nor does it rely on defining an effective field over a finite frequency range. Instead, the electro-optic coefficient is obtained directly from the phonon equation of motion, using the experimentally measured THz electric field as input.

2. Peak position of SHG modulation spectrum vs. peak of phonon resonance $\varepsilon_2(\omega)$:

“In fig. 1d and f and throughout the text, the authors seemed to believe that the spectrum of modulated SHG signal has a peak at exact the phonon frequency of 4.3 THz. However, at the exact resonance the penetration depth of the THz field could be very limited (perhaps $\sim 1 \mu\text{m}$ since the absorption coefficient is normally on the order of 10^4 cm^{-1} at resonance). And above the resonance, contribution from the phonon mode will go negative to the EO effect. Actually, it looks like that the peak position of the SHG spectrum (Fig. d and f) is slightly below that of the peak in ε_2 , please confirm that.”

Indeed, a direct comparison between the peak of the SHG modulation dynamics and the phonon frequency should be made. The spectrum of the SHG modulation is at 4.22 THz, slightly below the phonon resonance at $\omega_Q = 4.32$ THz. As the reviewer suggests, the offset between the two peaks can be attributed to the frequency dependence of the vibrational electro-optic coefficient. This coefficient typically exhibits a dispersive behaviour: it reaches a positive maximum just below the phonon resonance and becomes negative immediately above it. To evaluate the frequency dependence of the electro-optic coefficient near the phonon resonance, we computed it using the following equation from Ref. [R1]:

$$r_{22}^E(\omega) = \frac{r_{22}^E(0)\omega_Q^2}{\omega_Q^2 - \omega^2 - i\Gamma\omega} \quad \text{Eq. R1}$$

Here $\omega_Q = 4.32$ THz is the phonon frequency, $\Gamma = 0.19$ THz is its spectral width measured by FTIR spectroscopy, and $r_{22}^E(0)$ is the vibrational contribution of the phonon to the electro-optic coefficient in the DC limit.

Fig. R1. **a**, Real and imaginary parts of $r_{22}^E(\omega)$ calculated from Eq. R1, shown in arbitrary units since $r_{22}^E(0)$ provides only an overall scaling factor. Vertical dashed line indicates the phonon resonance frequency $\omega_Q = 4.32$ THz. The inset provides a zoomed-in view near the resonance. **b**, Comparison between the $r_{22}^E(\omega)$ and the Fourier transform of the measured ΔI_{SHG} dynamics. **c**, Attenuation coefficient in the THz range calculated from optical functions from FTIR spectroscopy.

We assume that the vibrational electro-optic response is dominated by the E -symmetry phonon mode at 4.32 THz, which is supported by recent DFT calculations showing that the contribution from this mode

to the electro-optic coefficient is at least an order of magnitude greater than from any other phonon mode within the THz pump spectral range (see Ref. [R2]).

For a finite phonon linewidth, $r_{22}^E(\omega)$ is a complex quantity. The real part of r_{22}^E modulates the real part of the refractive index – thereby affecting the SHG signal through phase-matching as discussed in the manuscript and supplementary information. The real and imaginary parts of $r_{22}^E(\omega)$ calculated from Eq. R1 are shown in Fig. R1a. The real part of $r_{22}^E(\omega)$ reaches its positive maximum just below the phonon resonance frequency - indicated by the dashed line in Fig. R1a.

In Fig. R1b, the real component of $r_{22}^E(\omega)$ is compared with the Fourier spectrum of the experimentally observed THz induced SH modulation dynamics. The positive peak of the real part of $r_{22}^E(\omega)$ coincides with the maximum of the SH modulation. After the positive peak, the Fourier amplitude of SHG modulation dynamics drops sharply, consistent with the onset of strong absorption above the phonon resonance.

The electric field attenuation coefficient, defined as $1/\alpha = 2\pi\nu_{\text{THz}}k/c$ is calculated from the imaginary part of the refractive index k (determined experimentally with FTIR spectroscopy) is shown in Fig. R1c. At the phonon resonance frequency of 4.32 THz, the electric field attenuation depth α is approximately 2 μm – as also noted by the referee. In contrast, for frequencies below ~ 4.2 THz, the attenuation depth exceeds 30 μm . At these lower frequencies (below 4.2 THz), the THz–NIR interaction length of the THz EO modulation process, is also on the order of 30 μm . It is given by $lc = \frac{c}{2\nu_{\text{THz}}|n_{gr} - n_{\text{THz}}|}$, where $n_{gr} = 1.68$ is the NIR group index at 800 nm and $n_{\text{THz}} \sim 2.9$ is the THz phase index, as discussed in Ref. [R3]. (n_{THz} is shown in Fig. R2a.)

The frequency dependence of the electro-optic coefficient is now discussed in detail in the revised manuscript. Fig. 5 in the revised manuscript now includes the frequency dependence of the real part of the electro-optic coefficient alongside the SHG modulation spectrum for direct comparison. In addition, we have added a new section to the Supplementary Information (Sec. S9) that expands on this point.

3. Determination of phonon frequencies and spectral weights:

“Related to 2, the $\epsilon\epsilon_2$ curve and its peak positions were obtained by fitting the FTIR absorption spectrum of thin BBO crystal (Fig. S3). However, from 4 to 8 THz the transmittance is practically zero. How much reliable will be the phonon frequencies spectral weights determined this way in that range (e.g. 4.3 THz-144.83 cm^{-1} , and 188.78, 245.15 cm^{-1}).”

The THz transmission is reduced near the phonon resonance; however, we do not expect this to compromise the accuracy of the measured phonon frequencies and linewidths. These parameters were determined using a Kramers–Kronig-consistent fitting procedure implemented in the Reffit software [R4], which allows reliable extraction of the optical functions even in regions of low transmission. The fitting was initialized using phonon frequencies reported in the literature (Refs. [R6,R7]).

The results of our fits for the real and the imaginary part of the refractive index along the ordinary axis are shown in Fig. R2a. The vertical grey lines in the figure correspond to the retrieved phonon frequencies at $\omega_{Q1} = 4.32$ THz, $\omega_{Q2} = 5.71$ THz, $\omega_{Q3} = 7.34$ THz.

For comparison, we have also plotted the results of a similar measurement reported in Ref. [R5] (Fig. R3b). Here the phonons were reported at $\omega_{Q1} = 4.65$ THz, $\omega_{Q2} = 5.65$ THz, $\omega_{Q3} = 7.34$ THz. There is a small discrepancy of 0.3THz at ω_{Q1} that might originate from sample quality and/or instrumental resolution. Our experiments were performed on a 30 μm -thick *uncoated* high-crystal-quality BBO sample and the transmission was evaluated using FTIR spectroscopy with a resolution of 0.014 THz. The measurement reported in Ref. [R6] was performed with a 400 nm anti-reflection coated BBO crystal that could affect the THz transmission and the transmission was evaluated using THz-TDS with

a lower frequency resolution (0.05 THz). These factors could explain a smearing effect in the refractive index, which is also discussed in Ref. [R7].

Additional evidence supporting the validity of the phonon frequencies measured in our work is provided by Raman spectroscopy, which yields E -symmetry phonon modes at 143 cm^{-1} , 187 cm^{-1} and 245 cm^{-1} , see Ref. [R8]. These values are nearly identical to ours: $\omega_{Q1} = 4.32\text{ THz} \rightarrow 144\text{ cm}^{-1}$; $\omega_{Q2} = 5.65\text{ THz} \rightarrow 189\text{ cm}^{-1}$; $\omega_{Q3} = 7.34\text{ THz} \rightarrow 245\text{ cm}^{-1}$.

Fig. R3. a, Imaginary part of the refractive index k and real part n (inset) along the ordinary axis obtained by Kramers–Kronig-consistent fitting procedure from our transmission spectra measured by FTIR spectroscopy. Vertical lines correspond to phonon frequencies of $\omega_{Q1} = 4.32\text{ THz}$, $\omega_{Q2} = 5.65\text{ THz}$, $\omega_{Q3} = 7.34\text{ THz}$. **b**, Refractive index n and k extracted from Ref. [R6].

4. Determination of the refractive index with DFT:

“In the text, line 268, the refractive indices by DFT calculation were given as $n_x = n_y = 1.705$, yet in the in line 374, $n_x = 1.76$ and $n_y = 1.76$ were listed. At what wavelength of those values are referred to, it is about 1.66 at 800 nm from measured values.”

Indeed, this is a typo. The correct index of refraction, is $n_x = n_y = 1.705$.

This value was obtained using density-functional perturbation theory (DFPT), see Ref. [R9], which yields the macroscopic dielectric tensor in the long-wavelength limit ($q=0$) whose square-root gives the index of refraction. The long-wavelength approximation is valid in weakly dispersive transparent materials, as is the case for BBO in the optical region of the spectrum (Ref. [R10]).

Our values are consistent with previous DFT studies reporting a refractive index of $n_x = n_y = 1.692$ at 800 nm, see Ref. [11]. The small discrepancy could arise from the fact that we also included local-field effects (beyond the independent-particle approximation employed in Ref. [R11]).

Experimentally, the refractive index is known to be 1.66 at 800nm (Ref. [R10]). The agreement between our DFT calculation of the index of refraction and the measured value serves to validate our calculations, which is important because the same approach is used to evaluate dynamical changes in the refractive index due to phonon-driven atomic displacements.

It should be noted, that the experimental value is used in the analytical modelling.

The typo has been corrected in the manuscript and additional computational details and comparison with the experimental refractive index have been incorporated in the Methods section.

5. Estimation of the electro-optic coefficient in SI unit:

“The vibrational electro-optic coefficient r_{22} was estimated to be $0.0014 \text{ \AA}^{-1} (\text{amu})^{-1/2}$, which is not in the standard unit. It must be converted to the SI unit (m/V) for comparison to the electronic contributions or the contributions from all the phonons at off-resonance frequencies.”

Analytically, the modulation of the refractive index Δn is related to the phonon coordinate Q by

$$\Delta n = \frac{n_{x'}^3 r_{22} Q}{2}, \quad \text{Eq. R2}$$

see Eq.2 in the main manuscript. Here, r_{22} denotes the phonon–optic coupling coefficient that quantifies the coupling between the phonon amplitude and the resulting refractive-index modulation.

From the DFPT results, Δn varies linearly with Q by which a value of $r_{22} = 0.0049 \text{ \AA}^{-1} (\text{amu})^{1/2}$ can be extrapolated from a linear fit, as discussed in the main text.

The standard vibrational EO coefficient, which is denoted here as r_{22}^E describes the variation of the index of refraction with respect to the applied electric field strength. This coefficient can be given in SI units (pm/V).

The conventional linear relationship between the phonon coordinate Q and the electric field E_{THz} :

$Q(\omega) = \frac{\sqrt{\epsilon_0 V} \omega_p}{(\omega_Q^2 - \omega^2 - i\Gamma\omega)} E_{THz}(\omega)$ is required to convert between the two coefficients. Substituting this relationship into Eq. R2 yields

$$\Delta n = \frac{n_{x'}^3}{2} r_{22} \frac{\sqrt{\epsilon_0 V} \omega_p}{(\omega_Q^2 - \omega^2 - i\Gamma\omega)} E_{THz} = \frac{n_{x'}^3}{2} \tilde{r}_{22}^E E_{THz} \quad \text{Eq. R3}$$

Here, V is the volume of the unit cell, ϵ_0 is the vacuum permittivity, ω_Q , ω_p and Γ are the phonon frequency, phonon plasma frequency and the linewidth, respectively. Here, \tilde{r}_{22}^E is the complex EO coefficient in SI units, and taking its real part:

$$r_{22}^E = \text{Re}\{\tilde{r}_{22}^E\} = r_{22} \frac{\sqrt{\epsilon_0 V} \omega_p (\omega_Q^2 - \omega^2)}{(\omega_Q^2 - \omega^2)^2 - (\Gamma\omega)^2} = r_{22}^E(0) \frac{(\omega_Q^2 - \omega^2)}{(\omega_Q^2 - \omega^2)^2 - (\Gamma\omega)^2}. \quad \text{Eq. R4}$$

Note that Eq. R4 corresponds to the usual expression of the vibrational EO coefficient (see e.g. Ref. [R1]), also shown in in Eq. R1 of this report.

Using $r_{22} = 0.0049 \text{ \AA}^{-1} (\text{amu})^{1/2}$ from the DFPT-derived refractive-index modulation, and $\omega_p = 6.57$ THz, $\omega_Q = 4.32$ THz and $\Gamma = 0.19$ THz determined by infrared spectroscopy, a maximum positive value of $r_{22}^E - \text{max} = 55 \frac{\text{pm}}{\text{V}}$ at $\omega_* = \sqrt{\omega_Q^2 - \Gamma\omega_Q} = 4.25$ THz is obtained. As discussed in the main text, the EO coefficient does not peak exactly at the phonon frequency ω_Q , the real part attains its maximum just below resonance at ω_* which is also consistent with the frequency peak of the SHG modulation response (see Fig. 5b of the main text).

A comparison of the calculated EO coefficient with experimental values can be performed in the DC limit. Extrapolating Eq. S48 to $\omega \rightarrow 0$ yields $r_{22}^E(0) = 6.6 \text{ pm/V}$, which is larger than the commonly reported experimental DC value of approximately 2.5 pm/V for BBO [R13]. This discrepancy likely reflects uncertainties in the experimental determination of the phonon parameters, as well as the limitations of the single-phonon DFPT approach in accurately describing the low-frequency electro-optic response, which includes both electronic contributions and the cumulative effect of multiple phonon modes.

Below, we provide an experimental estimate of the vibrational EO coefficient based on measurements of the SHG modulation dynamics, and from the phonon dynamics calculated using the measured THz electric field.

Experimental estimation of the EO coefficient from broadband THz pumping: Experimentally, the value of r_{22}^E can be estimated from the measured SHG modulation dynamics. As discussed in the main text, the relative SHG modulation, at fixed polarisation angle α is: $\Delta I_{rel}(t, \alpha) = \Delta I_{SH}(t, \alpha)/I_{SH,0}(\alpha)$, where $I_{SH,0}(\alpha) = I_{SH,0}(0) \cos^4 \alpha$. Measurements made at $\alpha = 30^\circ$ were used to evaluate the EO coefficient.

Analytically, the modulation of the SH dynamics in the time domain as a function of α (Eq. 2 in the main text) is:

$$\Delta I_{SH}(\alpha, t) = 4 I_{SH,0}(0) \gamma(t) \frac{\Lambda(\delta_{ph}, L)}{L} \cos^3 \alpha \sin \alpha, \quad \text{Eq. R5}$$

therefore

$$\Delta I_{rel}(\alpha, t) = 4 \gamma(t) \frac{\Lambda(\delta_{ph}, L) \sin \alpha}{L \cos \alpha} = 4\kappa(\theta) r_{22} Q(t) \frac{\Lambda(\delta_{ph}, L)}{L} \tan \alpha. \quad \text{Eq. R6}$$

Here, $\kappa(\theta = 29.3^\circ) = 65$ is the crystal geometrical factor, Λ is the parameter that accounts for the finite phonon penetration depth $\delta_{ph} = 30.2 \text{ }\mu\text{m}$, within a crystal of thickness $L = 300 \text{ }\mu\text{m}$ and $Q(t)$ is the time dependent phonon coordinate. $Q(t)$ can be calculated using the phonon equation of motion leaving r_{22} as the only free parameter to find agreement with the experimentally measured relative modulation.

The phonon dynamics are described by:

$$\ddot{Q} + \Gamma \dot{Q} + \omega_Q^2 Q = \sqrt{\epsilon_0 V} \omega_p \tilde{E}_{THz}(t), \quad \text{Eq. R7}$$

where ϵ_0 is the vacuum permittivity, $V = 596 \text{ \AA}^3$ is the unit cell volume, and $\omega_Q = 4.32 \text{ THz}$, $\omega_p = 6.57 \text{ THz}$ and $\Gamma = 0.19 \text{ THz}$ are the phonon frequency, phonon plasma frequency and the linewidth measured by THz-FTIR spectroscopy. $\tilde{E}_{THz}(t)$ is the THz field transmitted through the air-sample interface: $\tilde{E}_{THz}(t) = \frac{1}{2\pi \int \tilde{t}(\omega) E_{THz}(\omega) e^{-i\omega t} d\omega}$, where $\tilde{t}(\omega)$ is the complex Fresnel transmission calculated from the BBO refractive index (determined by THz FTIR), $E_{THz}(\omega)$ is the spectrum of the THz pump field measured by EOS.

Solving Eq. R7 for $Q(t)$ and substituting into Eq. R6, agreement between the calculated relative modulation and measurement is obtained for a phonon-optic coupling coefficient, $r_{22} = 0.0130 \text{ \AA}^{-1} (\text{amu})^{1/2}$ (Fig R4) with corresponding peak value of the EO coefficient, $r_{22}^E\text{-max} = 145 \frac{\text{pm}}{\text{V}}$.

Fig. R4. Experimentally measured ΔI_{rel} and calculated one using Eq. R6.

Estimation of EO coefficient from narrowband THz pumping: Broadband THz pumping contains spectral components far from the phonon resonance which may interfere with the determination of r_{22}^E . Therefore, an important cross-check is to evaluate r_{22}^E with a narrowband THz excitation using band-pass THz filters to isolate spectral components centered around the phonon frequency.

The optical setup was based on a 1-kHz repetition-rate Ti:sapphire laser. THz radiation was generated in a DSTM crystal and driven by OPA signal pulses with an energy of 0.44 mJ per pulse. The resulting THz waveform and the corresponding field-strength estimation were obtained via EOS using a 200- μm -thick GaP crystal. The THz electric field was determined using the conventional expression (see, e.g., Ref. [12]), including the Fresnel transmission coefficient. The measured temporal waveform and its Fourier amplitude spectrum are shown in Fig. R5a and Fig. R5b, respectively.

The relative SH modulation dynamics $\Delta I_{rel}(\alpha, t)$, were recorded at the polarization angle ($\alpha = 30^\circ$), and the corresponding Fourier amplitude spectrum is shown in Fig. R5c and Fig. R5d, respectively.

Following the same procedure as in the broadband THz case, we find excellent agreement between the measured and calculated time-domain dynamics (Fig. R5c) for a phonon–optic coupling coefficient of $r_{22} = 0.0160 \text{ \AA}^{-1} (\text{amu})^{1/2}$, which corresponds to $r_{22-max}^E = 178 \frac{pm}{V}$ - comparable to the value obtained using the broadband THz excitation.

The relatively minor discrepancy (less than a factor of 3) between theory and experiment, observed in both the broadband and narrowband cases, may originate from several sources. Near the phonon resonance, the electro-optic coefficient is strongly dispersive, and the associated experimental uncertainties are therefore expected to be significant. These uncertainties arise from the interplay of multiple factors, including accuracy in the determination of the optical constants, phonon parameters, and the effective strength of the applied “internal” THz driving field. On the other hand, the theoretical treatment may also introduce systematic deviations, as it adopts a simplified description that considers only a single phonon mode.

Importantly, overall agreement between the experimentally determined and theoretically predicted electro-optic coefficients validates the DFPT calculations, which enables further computational exploration of the interaction between phonon modes and the local electronic environment. These interactions are of broad interest in the materials community and currently an active area of research in nonlinear optics [Refs. R14,R15]

We have revised main manuscript, updated Fig. 5, and included a new section (Sec. S9) on the calculation of the EO coefficient in the Supplementary Information.

Fig. R5. SHG modulation with a narrowband THz pulse. *a*, Time-domain THz pump waveform measured by EOS. *b*, Amplitude spectra corresponding to *a*. *c*, THz-driven ΔI_{rel} and calculated one using Eq R6. *d*, Amplitude spectra corresponding to curves in *c*.

In conclusion, we believe that we have now carefully addressed all points in the report and would like to thank the reviewer for their time and consideration. The reviewer's constructive feedback and insight as proven extremely important in improving the clarity and rigor of the revised manuscript and Supplementary Information.

References Report 1:

- [R1] Casalbuoni, S. et al. "Numerical studies on the electro-optic detection of femtosecond electron bunches." *Physical Review Special Topics—Accelerators and Beams* 072802, 11 (2008).
- [R2] R. Li, "Vibrational contributions to the electro-optic effect of BBO, KTP and RTP crystals." *Computational Materials Science* 230, 112529 (2023).
- [R3] Valverde-Chávez, D. A., and Cooke G. D. "Multi-cycle terahertz emission from β -barium borate." *Journal of Infrared, Millimeter, and Terahertz Waves* 38.1, 96-103 (2017).
- [R4] Kuzmenko, A. B. "Kramers-Kronig constrained variational analysis of optical data", *Review of Scientific Instruments* 76, 083108 (2005).
- [R6] Liu, J., et al. "Optical property of beta barium borate in terahertz region." *Applied Physics Letters* 93.17 (2008).
- [R7] Li, R., et al. "Optical properties of barium borate crystal in the THz range revisited." *Optics Lettera* 50, 686-689 (2025).
- [R8] Ney, P., et al. "Assignment of the Raman lines in single crystal barium metaborate." *Journal of Physics: Condensed Matter* 10.3, 673 (1998).
- [R9] Baroni, S., et al. "Phonons and related crystal properties from density-functional perturbation theory." *Reviews of modern Physics* 73.2, 515 (2001).

- [R10] Li, R., et al. "Wide spectral range refractive indices of BBO crystal covering 0.2–2000 THz." *Applied Optics* 64.15, 4410-4414 (2025).
- [R11] Lin, J., Lee, M. H., Liu, Z. P., Chen, C., and Pickard, C. J. "Mechanism for linear and nonlinear optical effects in β -BaB₂O₄ crystals." *Physical Review B*, 60(19), 13380 (1999).
- [R12] Zhang, X.-C., and Jingzhou X.. *Introduction to THz wave photonics*. Vol. 29. New York: Springer, (2010).
- [R13] Salvestrini, J. P., Abarkan, M., and Fontana, M. D. "Comparative study of nonlinear optical crystals for electro-optic Q-switching of laser resonators." *Optical Materials*, 26(4), 449-458 (2004).
- [R14] Long-Qi, Y. et al. "Nonlinear Optical Mechanism of β -BaB₂O₄ Revealed by Experimental Electron Density." *Advanced Optical Materials* 12, 2301897 (2024).
- [R15] Cammarata, A., and Rondinelli, J. M. "Contributions of correlated acentric atomic displacements to the nonlinear second harmonic generation and response." *ACS photonics* 1, 96-100 (2014).

Reply to review report 2:

We thank the reviewer for carefully reading our manuscript and providing constructive criticism.

As the reviewer notes, phononic manipulation of material properties, especially in cuprates and ferroelectrics is of current interest. In those studies, the dynamics are observed using SHG as a probe and interpreted within the framework of THz Field Induced Second Harmonic generation (TFISH), which can be viewed as a time-dependent $\chi^2(t)$ or, alternatively, a $\chi^{(3)}$ -process.

At the same time, the reviewer notes that exploiting phonon-driven dynamics specifically for the purpose of SHG modulation and other nonlinear optics applications has not yet been systemically investigated or discussed in the literature. Our study in BBO was designed precisely for this purpose.

In general, however, the reviewer questions the interpretation of our results, suggesting that the observed effects are due to the familiar $\chi^{(3)}$ process, as was the case in the previous work involving TFISH. Indeed, we considered this point carefully, however, based on our analysis we were able to exclude $\chi^{(3)}$ processes as the main driver of the dynamics while simultaneously confirming the role of the cascaded $\chi^{(2)}$ process as the primary driver.

In the previous work, where SHG was used as a probe, the SHG process was *phase-mismatched* and frequency conversion is confined to a coherence length that is typically on the order of several microns [R16-R19]. Here, THz-driven phonons that directly modulate the second-order nonlinearity result in a dominant $\chi^{(3)}$ -effect.

In contrast, our experiments were performed in a BBO crystal optimised for Type-I phase matching at equilibrium, with a coherence length exceeding 300 μm . Here, although $\chi^{(3)}$ -effects also occur, it is the phonon-driven, electro-optic perturbation of the refractive-index ellipsoid that strongly affects the SHG efficiency via the phase-matching parameter Δk , which dominates the observed dynamics.

A dominant $\chi^{(2)}$ -effect can be expected, as BBO exhibits a strong phonon-mediated EO response [R20]. Here, driving the phonon perturbs the refractive-index ellipsoid (a 2nd-order effect), which in turn rotates the principal axes affecting the angle between the ordinary axis and the fundamental in the usual 2nd-order Type-I SHG process. In the phase matched geometry, this cascaded 2nd-order effect qualitatively and quantitatively describe the observed angular symmetry, linear electric field scaling, and the absolute modulation depth in the dynamic SHG signal.

Furthermore, from the measured response, an estimation of the vibrational electro-optic coefficient can be obtained, which is found to be in reasonable agreement with that extrapolated from the DC (Pockels) regime. In fact, electric-field-induced refractive-index control of phase-matched SHG in bulk nonlinear crystals is well established in the DC electro-optic ($\chi^{(2)}$) regime [R21,R22], and is conceptually distinct from EFISH (a $\chi^{(3)}$ -mediated SH process typically exploited in thin-films, few-layer materials or interfacial systems with strong bias fields). Our work extends the cascaded $\chi^{(2)}$ -mechanism to the THz phonon-driven domain, where lattice dynamics enable highly efficient control of phase-matched SHG.

Nevertheless, to isolate and demonstrate that $\chi^{(3)}$ effects are negligible in our type-I phase-matched experiment, we have now made additional measurements in BBO that has been intentionally detuned from the Type-I phase matching geometry. Here, with a coherence length on the order of a few microns, the $\chi^{(3)}$ -effect is in fact dominant, and the cascaded $\chi^{(2)}$ effect becomes negligible.

To summarise the outcome of these experiments, the equilibrium polarisation-dependent SHG signal and dynamic polarisation-dependent SHG modulation (at a local maximum of the effect in time-delay) in type-I phase-matched and *phase-mismatched* geometries is shown below in Fig. R6. Strikingly different angular symmetries observed in the phase-matched case in comparison to the phase-

mismatched case is the key evidence that reveals the origin of the dynamics as either a $\chi^{(2)}$ - or $\chi^{(3)}$ -effect, respectively.

Fig. R6. Isolating the TFISH contribution to THz-driven SHG modulation. *a*, Off-phase-matched geometry: the NIR and THz pulses propagate along the crystal c axis; SHG is collected along y . *b*, Phase-matched SHG geometry. *c*, Static “off-phase-matched” SHG: experimental data and fit curve with the function: $I_{SHG} \propto |d_{22}E_x^2 - d_{22}E_y^2|^2$. *d*, Static phase-matched SHG: experimental data with fit: $I_{SHG} \propto |d_{22}E_y^2|^2$, where the E_x^2 is not present because not phase-matched. *e*, THz-induced SHG modulation in the off-phase-matched case: data and fit: $\Delta I_{SHG} \propto a_1E_x^4 + a_2E_y^4 + a_3E_x^2E_y^2 + a_4E_x^3E_y + a_5E_xE_y^3$, modelling TFISH-driven modulation of the $\chi^{(2)}$ tensor components, see Ref. [16]. The a_3 is dominant and produces lobes oriented at 45° ; a_4 and a_5 are over one order of magnitude smaller. The maximum SHG intensity modulation is below 1% at the maximum THz field. *f*, THz-induced SHG modulation under phase-matched conditions: the angular symmetry differs qualitatively from *e* and cannot be described by TFISH alone.

In the phase-mismatched case:

- At equilibrium, 4-fold symmetry is observed in the SHG signal along the quadrant axes
- In the dynamical case, the original 4-fold symmetry is preserved, with the signal uniformly increasing or decreasing depending on the temporal overlap and sign of the THz field
- An additional 4-fold symmetric signal is observed along the diagonals, and this feature also oscillates uniformly with the THz field.

In the phase-matched case:

- At equilibrium, 2-fold symmetry is observed in the SHG signal along the vertical axis
- In the dynamical case, the 2-fold symmetry is superseded with 4-fold symmetry. The 4-fold symmetric lobes are aligned along 30° , 150° , 210° , 330° (not along the diagonals)
- The 4-fold symmetric signal does not oscillate uniformly with the driving THz field, rather pairs of opposing lobes increase or decrease in strength out of phase with each other.

The 4-fold symmetric signal in the phase-matched case at 30° is absent in the experiments isolating the $\chi^{(3)}$ effect. Further, in the phase-mismatched case, the maximum of the $\chi^{(3)}$ effect on the x - and y -axes

is absent in the phase-matched experiments. Finally, uniform increase or decrease of the 4-fold symmetric signal in the $\chi^{(3)}$ experiment is absent in the phase-matched geometry.

The phase-mismatched experiments are now referenced in the main manuscript, and a detailed explanation of the results are provided in the revised Supplementary Information (Sec. 7 and Sec. 8). A summary of that discussion is also included below in the detailed point-by-point response to the specific questions raised in the referee report. We thank the referee for the constructive review, which we believe has led to a substantially improved manuscript.

Response to specific criticisms:

Q1. Phonon symmetry: *“The material is not centrosymmetric, hence the mode cannot be of E_g symmetry.”*

The use of the E_g nomenclature in the manuscript was indeed a typographical error. The phonon is E symmetric as detailed in Ref. [R20] (also cited in our manuscript). The notation has been corrected.

Q2. Dielectric function and loss function:

“The phonon mode being driven is being displaced by the electric field. Hence, the relevant quantity is a Polarization P , which is proportional to χ and hence to ϵ_1 (not ϵ_2). The figure compares the response to ϵ_2 , which is a measure of coherent dissipation, and not of the induced phonon dipole. They should plot the loss function of the material ($1/\text{Im}(\epsilon_1)$) and compare the observed oscillation frequencies to existing phonons. Please correct accordingly.”

As pointed out in the report –the real part of the dielectric function $\epsilon_1(\omega)$ is proportional to the induced phonon dipole moment, therefore, the SHG modulation dynamics should be compared to $\epsilon_1(\omega)$ and not to the imaginary part $\epsilon_2(\omega)$. For reference, both $\epsilon_1(\omega)$ and $\epsilon_2(\omega)$ are plotted in Fig. R7a. Upon comparison with $\epsilon_1(\omega)$, we confirm that the SHG dynamics occur at a frequency that corresponds to the maximum of $\epsilon_1(\omega)$ and not at the frequency of the phonon mode given by the peak of $\epsilon_2(\omega)$. This point is considered in detail in our response to referee #1.

Fig. R7. a, Real and imaginary parts of the dielectric function, $\epsilon_1(\omega)$ and $\epsilon_2(\omega)$, as obtained from FTIR spectroscopy. Vertical dashed and solid black lines indicate, respectively, the frequencies where ϵ_1 reaches its positive maximum and ϵ_2 reaches its peak. **b,** The energy-loss function, $\text{Im}[-1/\tilde{\epsilon}(\omega)]$, plotted versus frequency. As can be seen, the maxima of ϵ_1 and ϵ_2 correspond to minima of the loss function

In addition to plotting $\varepsilon_1(\omega)$, reviewer #2 also requested that we calculate the loss function. We have now calculated the loss function ($\text{Im}[-1/\varepsilon(\omega)]$, Ref. [R23]) and observe that the phonon resonance is located at the minimum as shown in Fig. R7b.

Fig. 1 in the revised manuscript now includes both $\varepsilon_1(\omega)$ and $\varepsilon_2(\omega)$. We have still chosen to show $\varepsilon_2(\omega)$ in Fig. 1 of the manuscript for completeness, as its peak corresponds to the phonon frequency.

Q3. Homodyne and heterodyne SHG:

“More importantly. What are the authors measuring in their experiment? There are at least two contributions to the second harmonic signal when modulated by an IR phonon QIR. One contribution, which is a homodyned response, will be proportional to the square of the SH field and therefore one expects oscillations at twice the fundamental frequency. There will also be a heterodyne contribution, which will presumably rely on the interference between static SHG from the BBO crystal and the induced lattice oscillations. Have the authors studied the effect in detail and are they sure which made they are looking at? Oscillations at 4.3 THz could be a homodyned SHG from a 2.15 THz phonon or heterodyne oscillations from a 4.3 THz phonon. The best way to solve this issue would be to add a local oscillator to make sure that the heterodyned component dominates (see for example Mankowsky et al. PRL 118, 197601 (2017))”

As previously discussed, the reviewer questions whether the observed THz-induced SHG modulation arises from a 2nd- or 3rd-order TFISH effect, and if it is a 3rd-order effect, whether it is due to the **homodyne** or **heterodyne** term. As previously discussed, largely due to symmetry arguments, which are supported by the additional experiments we performed in a phase-mismatched geometry, the SHG modulation observed in our Type-I phase-matched experiments can be attributed to a cascaded 2nd-order effect and 3rd-order TFISH, or $\chi^{(3)}$ -effects can be excluded.

In the TFISH framework, the intensity of the SHG signal is proportional to three terms, a time-dependent heterodyne term, a time dependent homodyne term and a DC offset term [R24]:

$$I_{SHG}(t) \propto \left| \left(\chi^{(2)} + \chi^{(3)} E_{THz}(t) \right) E_{FF}^2 \right|^2 \quad \text{Eq. R9}$$

$$\propto \underbrace{\left| \chi^{(2)} E_{FF}^2 \right|^2}_{DC} + \underbrace{2 \text{Re}(\chi^{(2)} \chi^{(3)} E_{THz}(t) E_{FF}^4)}_{\text{Heterodyne} \propto E_{THz}(t)} + \underbrace{\left| \chi^{(3)} E_{THz}(t) E_{FF}^2 \right|^2}_{\text{Homodyne} \propto [E_{THz}(t)]^2}$$

To link this expression to optically driven phonons in the BBO, according to Ref. [R25], $\chi^{(3)} E_{THz}(t)$ is identical to a time-dependent perturbation of the 2nd-order non-linearity and can be rewritten in terms of the effective nonlinear coefficient d_{eff} and the phonon displacement Q :

$$\chi^{(3)} E_{THz}(t) = 2 \frac{\partial d_{eff}}{\partial Q} Q(t) \rightarrow \Delta d = \frac{\partial d_{eff}}{\partial Q} Q$$

1. Homodyne Component: Apparently, one concern of the referee here is that the observed SHG modulation arises from the **homodyne** response, i.e. the term $\propto (\chi^{(3)} E_{THz})^2$, which is in turn proportional to Q^2 . In this case, the SHG signal, observed to be modulated at 4.3 THz, would be driven by a phonon with half the frequency at 2.15 THz. Our measurements exclude this possibility:

- 1) **THz pump spectral selectivity.** Low-pass filtering the THz pump with a 4-THz cutoff (removing pump frequency components >4 THz; Fig. 2a–b of the main manuscript) nearly extinguishes the SHG modulation dynamics, even though the 2.15-THz pump component remains. Thus, excitation of the 2.15-THz mode does not result in SHG modulation. Rather, the

modulation requires spectral content near 4.3 THz, which is incompatible with a 2Ω (homodyne) signal arising from a 2.15-THz phonon.

- 2) **THz pump field scaling.** The SHG modulation amplitude scales linearly with the THz driving field (Fig. 2c-d of the main manuscript), consistent with a linear term $\propto Q \propto E_{THz}$. A homodyne term instead would scale as E_{THz}^2 .
- 3) **THz pump phase inversion.** Flipping the THz-field phase by 180° produces a π -phase inversion of the SHG modulation, see Fig. 2c of the main manuscript, as expected for a linear $\propto Q$ contribution; a quadratic $\propto Q^2$ would be invariant under this operation.

2. Heterodyne component: Without a systematic investigation of the polarisation dependence, the heterodyne term in Eq. R9 might be erroneously identified to be the origin of the SH modulation. The heterodyne term is in fact linearly dependent in strength on the THz electric field, it is phase-locked to the THz electric field, and it has a frequency dependence that matches the THz-driven phonon. Nevertheless, the magnitude of the 3rd-order TFISH effect is expected to be much weaker than the 2nd-order effect. Moreover, when the angular symmetry of the intensity modulation is taken into account, it becomes evident that 3rd-order processes cannot be responsible for the observed dynamics.

If the x -component in the lab frame (corresponding to the extraordinary axis in the phase matched geometry) is isolated with a polariser, and for generality if all elements in the time-dependent $\chi^{(2)}$ tensor, or equivalently $\chi^{(3)}$ tensor, are assumed to be non-zero, the polarisation dependent intensity difference includes the following terms ($E_z = 0$):

$$\Delta I_{SH,THz}^x(\alpha) = I_{SH,THz}^x - I_{SH,0}^x \propto (a_1)E_x^4 + (a_2)E_y^4 + (a_3)E_x^2E_y^2 + (a_4)E_x^3E_y + (a_5)E_xE_y^3, \quad (R10)$$

Where:

$$E_x = E_0^{FF} \sin \alpha \quad ; \quad E_y = E_0^{FF} \cos \alpha \quad (R11)$$

In Eq. R10, the a_1, \dots, a_5 coefficients depend on the static and modulated nonlinear tensor elements. Given the initial symmetry of the BBO crystal ($d_{15}, d_{22}, d_{33} \neq 0$), $\Delta I_{SH,THz}^x(\alpha)$ has an angular dependence proportional to the terms in R10: $\cos^4 \alpha, \sin^4 \alpha, \cos^2 \alpha \sin^2 \alpha, \cos \alpha \sin^3 \alpha$ and $\cos^3 \alpha \sin \alpha$. This proportionality depends to first or second order on the time-dependent second order nonlinearity, or the 3rd-order susceptibility $\chi^{(3)}$.

The measured angular dependence of the SH intensity modulation in the phase-mismatched case is shown in Fig. R6e. Here, as previously stated, the TFISH, $\chi^{(3)}$ -effects are isolated and the angular dependence can be fit with an equation of the form in R10. The dominant fit parameters are found to be associated with the $\cos^4 \alpha, \sin^4 \alpha$ terms, which scale the lobes on the x - and y -axes, and the $\cos^2 \alpha \sin^2 \alpha$ term, which scales the lobes along the diagonals. Fine adjustment to the rotation of the polar fit can be made using the remaining terms. In the time-domain, the onset of the THz excitation results in a uniform decrease of the SHG signal on the axes, $\Delta I_{SHG} < 0$, which is consistent with a THz-induced decrease of the d_{22} coefficient. At the same time, the onset of the THz excitation results in a uniform increase of the SHG signal on the diagonals, $\Delta I_{SHG} > 0$.

The angular dependence in the Type-I phase-matched case instead is completely different, as shown in Fig. S5f. Here, the strongest variation in the SH intensity is found at $\pm 30^\circ$ and the signal on the x and y axes is minimised. Moreover, the sign of the effect alternates between lobes such that if one lobe experiences an increase in SH intensity, the neighbouring lobe exhibits a decrease. If the dynamics were explained by TFISH or, equivalently, $\chi^{(3)}$ -effects, one would expect that the $\cos^4 \alpha, \sin^4 \alpha$ and $\cos^2 \alpha \sin^2 \alpha$ terms would all play a role in the SH modulation, yet there is no clear signature of any of these terms in angular dependence of the phase-matched SH measurements. This is especially notable

for the $\sin^4 \alpha$ term, which involves a heterodyne term that mixes two Type-I phase matched processes, which should have the strongest response.

Instead, only the $\cos^3 \alpha \sin \alpha$ term from the TFISH scenario is present in the phase-matched angular dependence. A closer look at the elements in the proportionality coefficient of the $\cos^3 \alpha \sin \alpha$ term, reveals a heterodyne component that is linear in THz field strength and scales with $O(\Delta d)$ or $O(\chi_0^{(3)})$, and a homodyne component that is quadratic in THz field strength and scales with $O^2(\Delta d)$ or $O^2(\chi_0^{(3)})$.

Based on previous arguments, the homodyne term can be excluded. There are also reasons to suggest that the heterodyne term does not contribute significantly to the phase-matched intensity modulation either. Importantly, in Type-I phase matching geometry, the heterodyne interference term involves a SH component proportional to the nonlinear tensor d that results from the Type-I phase-matched process, but the other component, proportional to the time-dependent nonlinearity, results from a Type-II process, which is NOT phase matched.

DFT calculations of the phonon displacements can also be used to estimate the time-dependent 2nd-order nonlinear coefficients, Δd . By combining these coefficients with the phonon penetration depth δ_{ph} (calculated independently by DFPT), a simple two-layer model of the BBO crystal can be used to estimate the strength of the 3rd-order TFISH effects. This model consists of a uniformly pumped layer with thickness δ_{ph} a constant phonon amplitude Q_0 . The remainder of the crystal is considered to be at equilibrium.

The Type-I phase-matched component, which is proportional to $d_{22}^2 |E_y^{FF}|^4$ will be affected by the time-dependent term Δd_{22} , which is expected to have a maximum value of $\Delta d_{22} = 0.1 \text{ pm/V}$ with $d_{22} = 2.9 \text{ pm/V}$. In the 2-layer model which is expected to overestimate the strength of the effect, the relative 3rd-order intensity modulation of this component is calculated to be 0.68%, far below the modulation observed in the phase-matched experiment. Furthermore, in terms of the angular dependence, this modulation will result in a uniform scaling of the original equilibrium 2-fold symmetric pattern, which is not observed in the experiment.

The strength of the Type-I/Type-II heterodyne component can also be calculated using the other time-dependent nonlinear components, $\Delta d_{25}, \Delta d_{26}$, and Δd_{35} . With the same 2-layer model, the relative intensity modulation that can be expected from this term is approximately 0.3%. Even though this value is again expected to be an overestimate, it is still far below the modulation observed in the experiment.

Estimates of the strength of the $\chi^{(3)}$ -effect fail to support a TFISH explanation of the intensity modulation in the phase-matched experiment.

Neither the heterodyne nor the homodyne TFISH mechanisms explain the phonon-induced SHG modulation that we observed.

On the other hand, BBO has a strong electro-optic response—well known from its use in EO devices such as Q-switches. Additionally, recent theoretical work indicates that this electro-optic response is dominated by the infrared-active phonon at 4.32 THz [R20], leading to a resonant enhancement of the EO-effect when that mode is excited. Therefore, the electro-optic effect cannot be neglected, as in conventional TFISH experiments performed so far, but must be included when analysing the phase-matched SHG. In this regime, even small THz-phonon driven changes in the refractive indices modify the phase-matching condition and, consequently, the SH field that builds upon the propagation. This electro-optic pathway provides the natural explanation for the measured modulation.

In optimal Type-I phase matching, the SH signal is generated only from the projection of the FF beam along the ordinary axis. When the phonon is driven, electro-optical perturbation of the ellipsoid

effectively rotates the principal axes (n_o and n_e), see Fig. R8, and thus shifts the optimal phase-matching angle by $\gamma \propto Q$. This relatively small rotation translates into a large, efficient modulation of the SHG signal with a four lobes angular symmetry that differs from nonlinear coefficient modulation in TFISH experiments.

Crucially, not only the angular dependence is naturally reproduced, but also the linear $\propto Q$ scaling and quantitatively the modulation amplitude. From the measured SHG modulation, we can extract an electro-optic coefficient that is consistent with values extrapolated from the dc (Pockels) regime, reinforcing the assignment. Additionally, ab-initio calculations of the refractive indices and nonlinear coefficients further show that, in phase-matched BBO, this phonon-mediated electro-optic SHG modulation dominates over TFISH for any phonon displacement.

In short, our measurements cannot be explained by classical TFISH process; the results are strikingly and quantitatively reproduced by a resonantly enhanced electro-optic SHG modulation in phonon-driven regime.

We have revised the manuscript and the SI Sec. S8, to quantify the TFISH contribution.

Fig. R8: Conceptual explanation of THz-phonon-driven refractive-index modulation and its impact on phase-matched SHG: Static: Under type-I ($oo \rightarrow e$) phase matching, the SHG polar response follows, $I_{\text{SHG}} \propto \cos(\alpha)^4$, where α is the pump-polarization angle relative to the ordinary axis; the index-ellipsoid tensor is diagonal. **Driven:** THz driven phonon excitation induces off-diagonal elements in the ellipsoid tensor (proportional to the phonon amplitude Q), equivalent to a small rotation of the principal axes by $\gamma \propto Q$, and a dynamic shift of the phase-matching angle. **Predicted SHG modulation.** For small γ , the change in SHG is $\Delta I_{\text{SHG}}(\alpha) \propto \cos(\alpha + \gamma)^4 - \cos(\alpha)^4 \sim -\gamma \cos^3(\alpha) \sin(\alpha)$, yielding the characteristic four-lobe pattern with alternating signs, as confirmed experimentally in panel and linear scaling: $\Delta I_{\text{SHG}} \propto \gamma \propto Q$, as experimentally observed. Detailed theory is discussed in the manuscript.

Q4. Phonon Symmetry: “Results of figure 4. I was confused in reading about this part of the paper. If the mode is an E mode, it means that it is doubly degenerate and I imagine that there will be two sets of mode displacements (lets call them E1 and E2) which will respond to the THz drive. If I look at figure 1 I imagine that a second set of vibrations will be found at some angle. Can the authors clarify this aspect? Maybe this is already contained in their analysis and hence this simply needs to be made clearer in the text.”

The IR-active phonon excited by the THz field has E symmetry, and the crystal belongs to the $R3c$ space group. Phonon frequencies and displacement vectors were obtained from DFPT calculations. The

unit cell, shown in Fig. R8a, is represented in cartesian coordinates x' , y' (in-plane) and z' (out-of-plane) of the crystal frame.

The E -mode at 144 cm^{-1} is doubly degenerate, with two orthogonal dipole moment components in the x' - y' plane: E_1 with dipole moment along the x' -axis and the E_2 with dipole moment along the negative y' -axis, as illustrated in Fig. R9a. The x -axis corresponds to the crystallographic a -axis, which is also the optical ordinary axis in the laboratory frame.

In the experiment, the THz pump polarization is aligned along the a -axis (ordinary axis). As the two degenerate E -mode branches form an orthogonal basis in the x' - y' plane, the THz pump field couples only to E_1 along the a axis. The E_2 mode, whose dipole lies perpendicular to the field, is not directly excited. The orientation of the single crystal BBO used in the experiment was confirmed using Laue X-ray diffraction, with the resulting diffraction pattern shown in Fig. R9b.

We have improved the text clarifying the symmetry of the E -mode in the revised manuscript and provided details in Sec. 3 of the revised Supplementary Information.

Fig. R8. *a*, Crystal unit cell from DFT calculations, shown in Cartesian coordinates x' , y' (in-plane) and z' (out-of-plane). The x -axis corresponds to the crystallographic a -axis. The E -mode is doubly degenerate, with branches E_1 and E_2 having dipole moment directions indicated by the orange arrows. *b*, Laue diffraction pattern of the single crystal BBO sample ($\theta = 29.3^\circ$, $\phi = 90^\circ$) used in the experiment to verify the experimental geometry. The THz electric field is aligned along the a -axis. In the Laue measurement, the crystal was tilted by -29.2° to confirm the $R3c$ symmetry. Red dots indicate simulated Laue spots generated using Laue diffraction software.

Q5 Refractive index dispersion: “The authors should maybe mention that this effect is in fact a third order ($\chi^{(3)}$) effect, in which two photons from the mid infrared pulse are mixed with one photon from the THz modulator. Indeed, the modulated SHG will be upshifted and downshifted by one phonon, which may be important for the phase matching conditions. In this sense, the propagation may have to include polariton equations, since on and off resonant excitation of a phonon will lead to a modulating field that propagates at vastly different group and phase velocities.”

If the dynamics were dominated by a $\chi^{(3)}$ effect, phonon mixing would shift the modulated SHG spectrum (up- or downshifted by one phonon) and could affect the propagation through dispersion or group-velocity mismatch. We want to emphasize again, that the dynamics in our experiment are not driven by a $\chi^{(3)}$ effect.

Nevertheless, in our BBO experiment the driven mode is at 4.32 THz, so the corresponding shift of the SH wavelength would be small: $\approx \pm 2$ nm at 400 nm. Over a ± 2 nm interval around 400 nm, the refractive index is effectively constant, as well as the phase and group velocities (see phase and group refractive index dispersions are show in Fig. R9). Therefore, the associated changes in phase mismatch, walk-off, and group-velocity mismatch induced by pumping the phonon are negligible. Conversely, in prior TFISH studies that drove much higher-frequency, mid-IR phonons (e.g., ~ 19 THz in LiNbO₃; Ref. [R19]) dispersion effects and the high frequency of the phonon involved are indeed important.

Fig. R9. Group and phase refractive indices $n_{e,g}$ and $n_{e,ph}$ along the extraordinary axis in the SHG wavelength region, calculated using the Sellmeier equation for the BBO crystal cut used in the experiment $\theta = 29^\circ$ and $\theta = 29^\circ$. The vertical dashed grey lines indicate the spectral bandwidth of the generated second harmonic (SH) in our experiment. The green dashed lines show the up- and down-shifted center wavelength corresponding to the addition and subtraction of the 4.35 THz phonon frequency.

In summary, our interpretation that the SHG modulation is based on a cascaded $\chi^{(2)}$ process can be used to entirely explain the observed angular symmetry, magnitude, and scaling with THz field strength. At the time of submission, we had excluded TFISH from playing a dominant role through analytical consideration of the angular symmetry and quantification of the strength of the third-order effects. Based on the reviewer's report and questions, we have expanded this treatment by explicitly considering the heterodyne and homodyne components in a potential TFISH signal. We have now also performed additional reference experiments in phase-mismatched BBO to isolate the $\chi^{(3)}$ effects. Comparison of these polarisation-resolved measurements with those made in phase-matched BBO allow us to definitively exclude TFISH as the mechanism behind the SHG modulation observed in our original experiments.

We are confident that we have addressed in this response, in the revised manuscript and the supplementary information also the other specific criticisms raised by the Referee .

Reference report 2:

[R16] Itoh, H., et al. "Terahertz Field Control of Electronic-Ferroelectric Anisotropy at Room Temperature in LuFe₂O₄." *Physical Review Letters* 135.10, 106504 (2025).

[R17] Bodrov, S. B., et al. "Terahertz-field-induced second optical harmonic generation from Si (111) surface." *Physical Review B* 105.3, 035306 (2022).

- [R18] Grishunin, K. A., et al. "THz electric field-induced second harmonic generation in inorganic ferroelectric." *Scientific Reports* 7.1, 687 (2017).
- [R19] Mankowsky, R., et al. "Ultrafast reversal of the ferroelectric polarization." *Physical review letters* 118.19, 197601 (2017).
- [R20] Li, R. "Vibrational contributions to the electro-optic effect of BBO, KTP and RTP crystals." *Computational Materials Science* 230, 112529 (2023).
- [R21] Konforty, N., et al. "Second harmonic generation and nonlinear frequency conversion in photonic time-crystals." *Light: Science & Applications* 14.1, 152 (2025).
- [R22] Schiek, R. "Nonlinear refraction caused by cascaded second-order nonlinearity in optical waveguide structures." *Journal of the Optical Society of America B* 10.10, 1848-1855 (1993).
- [R23] Zibold, A., et al. "Optical properties of single-crystal $\text{Sr}_2\text{CuO}_2\text{Cl}_2$." *Physical Review B* 53.1, 11734 (1996).
- [R24] Chia-Yeh, L. et al. "Broadband field-resolved terahertz detection via laser induced air plasma with controlled optical bias." *Optics Express* 23, 11436-11443 (2015).
- [R24] Fang, Y., Hao, J., Sang, J. et al. "Coherent manipulation of second-harmonic generation via terahertz-field mediated phonon-polariton in zinc oxide" *Nature Communications* 16, 5598 (2025).